# Enhancing deep neural networks through complex-valued representations and Kuramoto synchronization dynamics

**Sabine Muzellec**                                                  *sabine_muzellec@brown.edu*
*CerCo - CNRS, University of Toulouse, France*
*Carney Institute for Brain Science, Brown University, USA*

**Andrea Alamia**                                                          *andrea.alamia@cnrs.fr*
*CerCo - CNRS, University of Toulouse, France*

**Thomas Serre**                                                        *thomas_serre@brown.edu*
*Carney Institute for Brain Science, Brown University, USA*

**Rufin VanRullen**                                                    *rufin.vanrullen@cnrs.fr*
*CerCo - CNRS, University of Toulouse, France*

**Reviewed on OpenReview:** *https://openreview.net/forum?id=zx6QGmBL43*

## Abstract

Neural synchrony is hypothesized to play a crucial role in how the brain organizes visual scenes into structured representations, enabling the robust encoding of multiple objects within a scene. However, current deep learning models often struggle with object binding, limiting their ability to represent multiple objects effectively. Inspired by neuroscience, we investigate whether synchrony-based mechanisms can enhance object encoding in artificial models trained for visual categorization. Specifically, we combine complex-valued representations with Kuramoto dynamics to promote phase alignment, facilitating the grouping of features belonging to the same object. We evaluate two architectures employing synchrony: a feedforward model and a recurrent model with feedback connections to refine phase synchronization using top-down information. Both models outperform a real-valued baseline and complex-valued models without Kuramoto synchronization on tasks involving multi-object images, such as overlapping handwritten digits, noisy inputs, and out-of-distribution transformations. Our findings highlight the potential of synchrony-driven mechanisms to enhance deep learning models, improving their performance, robustness, and generalization in complex visual categorization tasks[*].

## 1 Introduction

Learning structured representations in artificial neural networks (ANNs) has been a topic of extensive research (Zhang et al., 2013; Chiou, 2022; Dittadi, 2023; Le Khac, 2024), yet it remains an open challenge (Schott et al., 2021; Dittadi, 2023). Notably, some researchers argue that the inability of ANNs to effectively bind and maintain structured representations may underlie their limited generalization capabilities and susceptibility to distributional shifts (Greff et al., 2020).

In neuroscience, the Binding Problem (Treisman, 1996; Roskies, 1999; Singer, 2007) refers to the brain's capacity to integrate various attributes of a stimulus—such as color, shape, motion, and location—into a unified perception. This process involves understanding how distinct features of an object are combined across different processing stages (in other words, object binding), enabling the brain to construct meaningful

---

[*]Code available at https://github.com/S4b1n3/KomplexNet

and cohesive representations of individual objects within a scene. Neural synchrony has been proposed as a key mechanism underlying this integrative process (Singer, 2007; Uhlhaas et al., 2009).

Kuramoto dynamics and related oscillator models have been widely studied (Budzinski et al., 2022; 2023) and specifically employed in computational neuroscience to explore synchronization phenomena in neural systems (Breakspear et al., 2010; Chauhan et al., 2022). These models provide insights into complex neural processes, such as phase synchronization and coordinated neural activity. Beyond neuroscience, the Kuramoto model has also found applications in artificial intelligence, offering a framework for understanding synchronization in complex systems (Ódor & Kelling, 2019; Rodrigues et al., 2016). Recently, its utility has extended to computer vision tasks (Ricci et al., 2021; Miyato et al., 2024; Benigno et al., 2023; Liboni et al., 2025), demonstrating its potential to enhance representation learning in ANNs.

Building on these insights, we propose leveraging the Kuramoto model to investigate the role of neural synchrony in convolutional neural networks (CNNs) for multi-object classification. We hypothesize that incorporating neural synchrony, inspired by the brain's solution to the Binding Problem, can be implemented using Kuramoto dynamics within ANNs, thereby enhancing their generalization abilities.

To test this hypothesis, we design a hierarchical model, KomplexNet, that integrates layers of complex-valued units with a bottom-up information flow. In KomplexNet, Kuramoto dynamics are applied at the initial layer to induce a synchronized state, which is then propagated through subsequent layers via carefully designed complex-valued operations. This approach enables the model to exploit the phase dimension of its neurons to bind visual features and organize visual scenes into distinct object representations, while the amplitude dimension retains standard CNN functionality.

We further extend KomplexNet by incorporating feedback connections to refine synchronization through top-down information (i.e., higher-level contextual signals, derived from global scene understanding or task goals, that influence local processing within the network). This extension demonstrates the critical role of top-down processes in enhancing phase synchrony and structuring object representations. Overall, our findings highlight the potential of neural synchrony mechanisms, modeled using Kuramoto dynamics, to improve the robustness, generalization, and representational capacity of deep learning architectures.

Overall, our contributions are as follows:

- We introduce KomplexNet, a complex-valued neural network that leverages Kuramoto dynamics for multi-object classification.

- KomplexNet has better classification accuracy than comparable baselines.

- KomplexNet also exhibits better robustness to images perturbed with Gaussian noise and generalization to out-of-distribution classification problems.

- Extending KomplexNet with feedback connections leads to better phase synchrony, exhibiting higher robustness and generalization abilities than KomplexNet without feedback.

## 2 Related work

**Complex-valued models.** Complex-valued neural networks are popular and extensively used in signal processing to model complex-valued data, such as spectrograms (see (Bassey et al., 2021) for a review). The term Complex-Valued Neural Network (CVNN) is commonly used to refer to *fully* complex networks: not only is the activation function complex, but so are the parameters. Trabelsi et al. (2017) proposes a list of operations adapted to a parametrization in the complex domain, including convolutions, activation functions, and normalizations. Moenning & Manandhar (2018) systematically compares complex-valued networks and their real-valued counterparts for object classification. Their findings highlight the importance of the choice of activation functions and architectures reflecting the interaction of the real and imaginary parts. However, when applied to real-valued data, the field lacks appropriate conversion mechanisms to the complex domain. One proposal by Yadav & Jerripothula (2023) includes a novel way to convert a real input image into the complex domain and a loss acting on both the magnitude and the phase. They implement their

transformation on several convolutional architectures and outperform their real-valued counterpart on visual categorization datasets. Additionally, such approaches often face challenges related to the non-differentiability of complex-valued loss functions and nonlinearities, which can complicate training and gradient propagation. In contrast, our model uses real-valued weights and applies complex-valued operations only in the activation space, allowing us to avoid these issues and train using standard backpropagation. Importantly, none of these papers use phase synchrony as a mechanism for perceptual organization in multi-object scenes.

**Synchrony in artificial models.** Some work has explored binding by synchrony or leveraging synchrony in artificial models without complex-valued activity. Ricci et al. (2021) proposes a framework for learning in oscillatory systems, harnessing synchrony for generalization. This approach is, however, limited by its learning procedure: the model is designed to learn to segment one half of an image and generalize on the other. Zheng et al. (2022) extends a spiking neural network with attention mechanisms to solve the binding problem, though the impact of synchrony on performance or robustness is not evaluated. While these models successfully group entities by synchronizing the spikes of neurons, synchrony was not designed to assist in solving visual tasks. In particular, it is unclear from the work if and how the resulting representations help improve the neural network's overall performance, robustness, or generalization ability.

**Complex-valued representations and binding by synchrony.** A growing body of literature uses complex-valued representations to explicitly model neural synchrony. Early models were designed to implement a form of binding by synchrony via complex-valued units to perform phase-based image segmentation (Zemel et al., 1995; Weber & Wermter, 2005) or object-based attention (Behrmann et al., 1998). These models were shallow architectures, and they were trained on small datasets. Different mechanisms, including feedback mechanisms (Rao et al., 2008; Rao & Cecchi, 2010; 2011), were later explored to influence synchrony in deeper architectures. However, all models were trained on datasets that remained limited to toy objects. Specifically, despite the simplicity of the objects, the images contained individual objects only, limiting the potential benefit of synchrony. Finally, Reichert & Serre (2013) scaled to Boltzmann machines and multi-object datasets. The authors proposed a general framework that included operations for binding by synchrony. Binding was shown to emerge through the phase of neurons. However, the approach did not include backpropagation training or end-to-end deep learning. Specifically, a real-valued Boltzmann machine was first trained, and the phases were introduced during test. Synchrony was, therefore, a completely emergent property and did not take any part in helping the model learn the task. Building on this work, Löwe et al. (2022) adapted the approach for training a complex auto-encoder to reconstruct multi-object images. This model was fully complex, even during training. The phase synchrony helped the model reconstruct an input image and outperform its real-valued counterpart. Finally, Stanić et al. (2023) scaled up the model to more objects and color images by adding a contrastive objective on the phases, followed by Gopalakrishnan et al. (2024) who improved the phase synchrony using recurrence and complex-weights. Our work distinguishes itself from previous work, notably in how we exploit synchrony mechanisms. In all the aforementioned approaches, synchrony is sought as an emergent property of the task and the neural operations implemented in the network. In contrast, we propose to introduce it using a Kuramoto system as an explicit synchronizer. We then study the benefit of synchrony for object categorization performance, as well as robustness and generalization.

## 3 Binding by Synchrony and Oscillatory Grouping Mechanisms

The binding by synchrony theory, as supported by a body of research both experimental (Singer, 2007; von der Malsburg, 1981) and computational Milner (1974); Grossberg (1976) (but see Roelfsema (2023); Shadlen & Movshon (1999) for alternative hypothesis), provides a comprehensive theory to understand how the brain integrates and perceives diverse sensory inputs. This theory asserts that synchronous neural activity, induced by neural oscillations, plays a foundational role in cognitive processes. According to this theory, when distinct features of a sensory stimulus are processed by specialized regions of the brain, the neurons responsible for representing these features synchronize their firing patterns at specific frequencies. This synchronization of neural oscillations enables precise coordination and the temporal binding of neuronal responses from different regions, thus uniting them into a coherent percept (Fries et al., 1997; 2002).

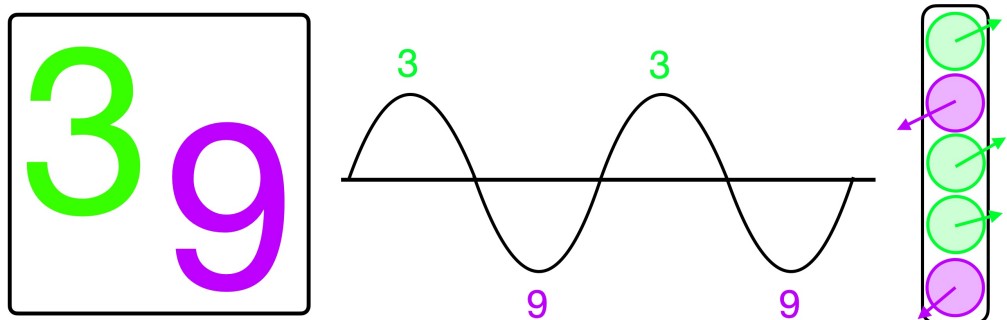

Figure 1: **The binding by synchrony hypothesis.** Brain activity exhibits an oscillatory pattern affecting a population of neurons; local neuronal interactions (e.g., excitation, inhibition) result in different groups of neurons being activated at different phase values. According to the binding by synchrony theory, neurons firing together (at the same phase) encode for the same object. Here, we use the phase of a complex number to represent this mechanism (right panel).

While earlier interpretations often linked this theory to Gestalt principles of perceptual organization – such as proximity and similarity (Wertheimer, 1938; Gray & Singer, 1989; Todorovic, 2008) – contemporary neuroscience has greatly expanded the mechanistic and empirical understanding of oscillatory dynamics. The study of neural oscillations has become central in systems neuroscience, providing insight into how different brain regions coordinate information processing across scales (Buzsaki & Draguhn, 2004). Oscillatory activity across frequency bands (e.g., theta, alpha, beta, gamma) has been shown to support attention, working memory, and multisensory integration. These rhythms are not isolated events but often participate in structured spatiotemporal patterns, such as traveling waves, which propagate information through cortical circuits (Jacobs et al., 2025; Keller & Welling, 2023; Mazzoni et al., 2015), and local field potentials (LFPs), which provide experimentally accessible signatures of population-level coordination (Haziza et al., 2024; Buzsáki et al., 2012).

Our model draws inspiration from this modern view: KomplexNet uses complex-valued representations to encode oscillatory activity and implements Kuramoto dynamics to coordinate phase alignment based on spatial proximity. While the model incorporates structural constraints reminiscent of Gestalt heuristics, it operates as a dynamical system in which synchrony is explicitly induced by structured coupling. We view synchrony as a controlled computational mechanism, guided by the architecture and coupling kernel, to group features based on spatial and task-relevant relationships. For simplicity, we assume a single oscillation frequency; the phase of each neuron reflects its alignment with this shared rhythm. Neurons with aligned phases are considered functionally grouped, reflecting coordinated patterns similar to those observed in structured neural activity during perception and cognition (see Figure 1).

In summary, while our model conceptually echoes early theories of binding and perceptual grouping, it is grounded in mechanisms that resonate with modern systems neuroscience. By aligning phase-based grouping with oscillatory principles observed in cortical circuits, such as LFP coherence, nested rhythms, and traveling waves, KomplexNet provides a tractable framework to explore how structured neural activity supports perceptual integration and cognitive organization.

## 4  KomplexNet: Kuramoto synchronized complex-valued network

In the following, we describe the details of KomplexNet's implementation. We use a Kuramoto system at the first layer $L_0$ to synchronize the phases and apply operations in the complex domain for subsequent layers, and we finish with the addition of feedback connections affecting the phase synchrony.

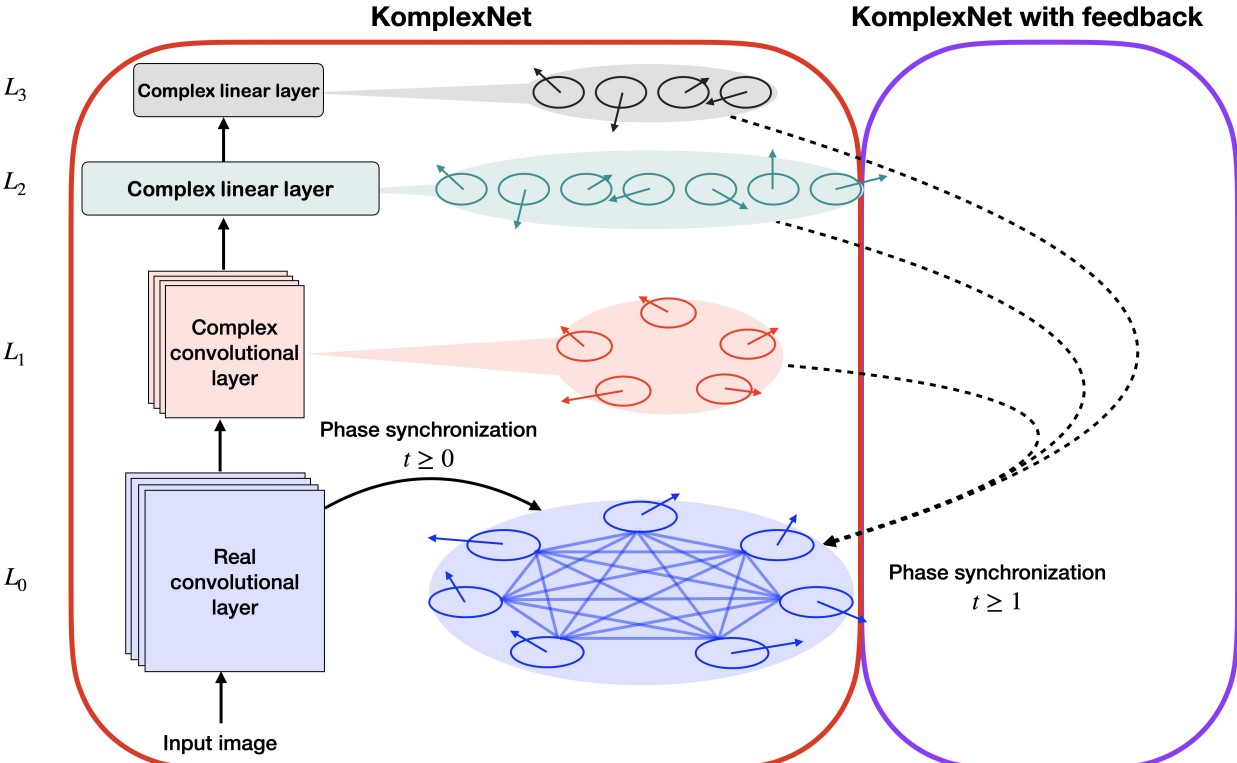

Figure 2: **Overview of KomplexNet.** We show the global architecture on the left and illustrate the phase dynamic on the right. The phases start with a random initialization and evolve with time according to Kuramoto's equation. The first complex representation results from the amplitudes and the phases of the first layer ($L_0$ in blue). The plain lines on the right represent the local connections inside the layer. The dashed lines represent some possible top-down connections to control synchronization. At every timestep, the phase of individual neurons gets updated by one Kuramoto iteration and propagated to the next network layers via complex-valued operations.

## 4.1 Kuramoto dynamic

**Kuramoto model.** Instead of considering synchrony as an emergent phenomenon, we choose to induce it using a Kuramoto system (Kuramoto, 1975) to organize the phases of the first layer. This process serves to synchronize the phases before incorporating them into the activity of the network. As we are dealing with complex-valued activity, we also need an amplitude value for each neuron at the first layer. This information is obtained by applying a real-valued convolution to the input image. To reach a synchronized state, we adapt the original equation of the Kuramoto model (Equation 3 in Kuramoto (1975)) and propose the dynamic described in Equation 1.

$$\dot{\theta}_{cij} = \eta \times [\sum_{k=0}^{C \times H \times W} (r_{k,cij} - \epsilon) \cdot \sin(\theta_k(t) - \theta_{cij}(t)) \cdot \tanh(a_k)] \tag{1}$$

The core idea of the Kuramoto model is to synchronize a population of oscillators by mutual influence. In our case, the oscillators are the phases $\theta \in \mathbb{R}^{C \times H \times W}$ (with H and W denoting the size of the input image), where a phase $\theta_{cij}$ will be influenced through the sine of the difference between itself and the rest of the population. This influence will be modulated by a learnable coupling kernel $r \in \mathbb{R}^{C \times C \times h \times w}$ (with $h \leq H$ and $w \leq W$) together with a global desynchronization interaction $\epsilon \in \mathbb{R}^1$, as well as the amplitude, $a$, associated to each influencing phase. In other words, each phase synchronizes with its neighbors (defined by the spatial range of the kernel) and desynchronizes with phases further apart. At the population level, the system favors

the emergence of several clusters of phases. Lastly, $\eta \in \mathbb{R}^1$ acts as a gain parameter, modulating the phase update at each timestep.

**Learning the coupling kernel.** The coupling kernel is expected to capture the interactions and mutual influence between phases. A positive/negative value in this kernel means that the corresponding two neurons will tend to synchronize/desynchronize their phases. The kernel should learn the inherent structure of the objects in the dataset to adjust the interactions between nearby phases. For example, in the case of handwritten digits, the objects are mostly vertical, hence the kernel should favor positive interactions between phases along the vertical axis. Here, we initialize the kernel with a 2D Gaussian to encourage interactions with closer neighbors.

To learn this kernel, we use the cluster synchrony loss defined in Ricci et al. (2021):

$$CSLoss(\theta) = \frac{1}{2}(\frac{1}{G}\sum_{l=1}^{G}V_l(\theta) + \frac{1}{2G}\Big|\sum_{l=1}^{G}e^{i\langle\theta\rangle_l}\Big|^2) \tag{2}$$

where $V(\theta)$ is the circular variance, $\langle\theta\rangle$ the average of a phase group, and $G$ the number of groups. The first part of this loss measures the intra-cluster synchrony while the second part represents the inter-cluster desynchrony. Minimizing the loss resolves to minimize the variance inside groups (each phase cluster should have the same value) and minimize the proximity between the centroids of the clusters on the unitary circle (the clusters should cancel each other out).

In Figure 3, we show some visualization of the phases obtained after 15 steps of our Kuramoto model on the Multi-MNIST dataset (Sabour et al., 2017) (see also Fig. 9 to visualize the learned kernel). The plot represents the 8 convolution channels, masked by the intensity of the amplitude, with the color indicating the phase value. The phases from the same digits are synchronized, and the two clusters of phases are desynchronized with an almost opposite position on the unitary circle. Interestingly, the model does not learn to systematically affect one specific phase value to a class of digits (as observable in Figure 3 with two '9' represented by distinct colors/phase values in the first two images). Indeed, in the `binding by synchrony` theory, there is no requirement that a given object should always and systematically be assigned the same phase value, as this would seriously limit the flexibility and adaptability of the coding system.

### 4.2 Complex representations in an artificial neural network

**Overall architecture.** Our model, KomplexNet combines the Kuramoto dynamic from Section 4.1 and additional complex operations described below, as represented in Figure 2 and Algorithm 1. To summarize, we start by extracting features from the input images by performing a real non-strided convolution (8 channels). The initial complex activity comprises these features along with random phases to perform one step of our Kuramoto dynamic. The resulting activation, $z_{L_0} \in \mathbb{C}^{8\times32\times32}$, is propagated in a bottom-up manner through one strided complex-convolution ($z_{L_1} \in \mathbb{C}^{8\times16\times16}$) and two linear layers (respectively $z_{L_2} \in \mathbb{C}^{50}$ and $z_{L_4} \in \mathbb{C}^{10}$). This is repeated for several timesteps to allow the phases to reach a stable synchronized state through the Kuramoto dynamic.

Each step of the Kuramoto model outputs a new state of the phases, combined with the amplitude extracted via the first real convolution to instantiate the complex activity.

**Complex operations.** We first redefine the standard operations of a convolutional network to be compatible with complex activations $z = m_z.e^{i\theta_z} \in \mathbb{C}$. We apply some biologically plausible transformations that allow the model to perform well, as used in (Reichert & Serre, 2013; Löwe et al., 2022; Stanić et al., 2023). We define the linear operations (convolutions and dense layers) as such:

$$z_1 = f_w(z) = f_w(Re(z)) + f_w(Im(z)).i \in \mathbb{C} \tag{3}$$

The newly obtained activity results from applying the same set of real-valued weights to both the real and imaginary parts of the input activity, modifying the amplitude and the phase jointly. Then, following the formulation first proposed by Reichert & Serre (2013), we apply a gating mechanism that selectively attenuates

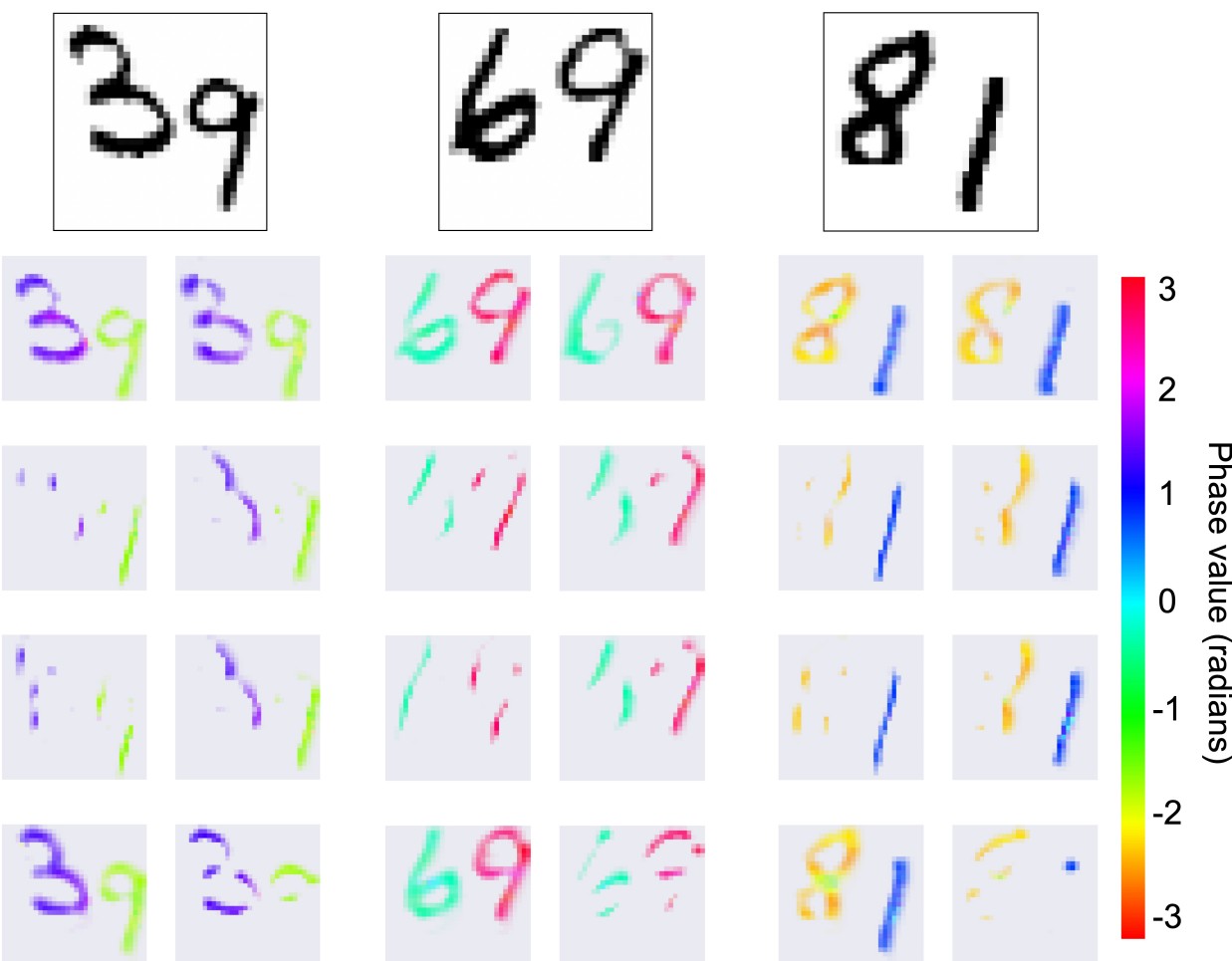

Figure 3: **Phase synchronization.** Given the input image shown at the top, we present visualizations of the phases from the first layer (across each of the 8 convolution channels) at the last timestep of the Kuramoto dynamic. The color represents the complex phase and we use the complex amplitude to mask out the background/non-active pixels.

inputs that are out of phase with the target activity. This prevents a scenario where desynchronized inputs (i.e., those with opposing phases) are treated equivalently to inhibitory inputs resulting from negative weights. Specifically, we compute a modulation term (named classic term in (Reichert & Serre, 2013)) $\chi$ as a function of the input amplitude, and we update the modulation variable $m_{z_2}$ by averaging the previous state $m_{z_1}$ with the newly computed term $\chi$.

$$\chi = f_w(|z|)$$
$$m_{z_2} = \frac{1}{2}(m_{z_1} + \chi) \tag{4}$$

This averaging serves to smooth the gating dynamics, allowing the network to gradually adapt modulation in response to changes in input amplitude, and ensuring that out-of-phase inputs do not undesirably mimic the effects of inhibition. Lastly, we apply a ReLU non-linearity only on the amplitude. Considering that a complex amplitude is by definition positive, we start by normalizing it before applying the desired function, as done by (Löwe et al., 2022; Stanić et al., 2023). This last step ends a block of operations representing one

layer in our model.

$$z_3 = ReLU(InstanceNorm(m_{z_2})).e^{i\theta_{z_1}} \in \mathbb{C} \tag{5}$$

We can additionally observe, in Figure 4 the effect of the complex operations by visualizing the phase distribution at each layer of the model at the last timestep. The first row represents the polar distribution of the color-coded phases at each layer. We can see that the Kuramoto model allowed the phases to reach a state with two opposite clusters – one for each digit – and this distribution is conserved in all the subsequent layers. We show in the second row the same activity but detailing the spatial information provided by the convolutions of the first and second layers.

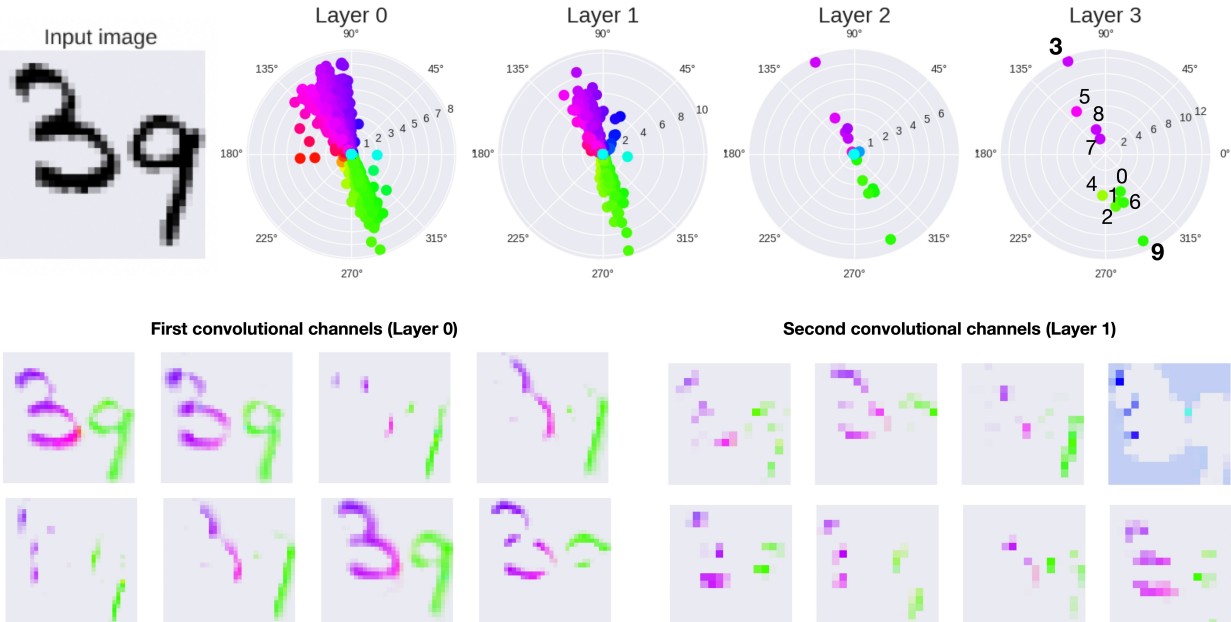

Figure 4: **Phases per layer.** Given an input image (top left corner), we show how the complex operations propagate the phase grouping instantiated at the first layer to the last decision layer (all activations are measured at the last step of the Kuramoto dynamic). As the first two layers ($L_0$ and $L_1$) are convolutional, we additionally visualize the eight convolutional channels in each layer (bottom).

### 4.3 Implementing feedback

We finally propose an extension of KomplexNet by implementing feedback connections to influence the phase synchronization (dashed lines on the right in Figure 2). At each timestep, the higher-level representations from the latter layers (carrying information about advanced features, object parts, and object classes) can help the synchronization process of the first layer. We combine the lateral synchronization Kuramoto dynamic of $L_0$ with feedback synchronization from higher layers. The resulting phase update per timestep at the first layer is described analytically in Equation 6. The first line defines $K_{cij}$ as the lateral synchrony in $L_0$ (same as Equation 1) and the second combines $K_{cij}$ with the sum of the Kuramoto dynamics across layers.

$$K_{cij} = \eta \times [\sum_{k=0}^{C \times H \times W} (r_{k,cij} - \epsilon) \cdot \sin(\theta_k(t) - \theta_{cij}(t)) \cdot \tanh(a_k)]$$

$$\dot{\theta}_{cij} = K_{cij} + \sum_{l=1}^{N_l}[\eta \times [\sum_{k=0}^{C_l \times D_l} (r_{lk,cij}) \cdot \sin(\theta_{lk}(t) - \theta_{cij}(t)) \cdot \tanh(a_{lk})]] \tag{6}$$

We instantiate the activity of each layer at the first timestep through a feedforward propagation and introduce the feedback starting at the second step. Algorithm 2 details the corresponding update in the dynamic of

the whole model. Similarly to the lateral coupling kernel of layer $L_0$, the feedback coupling kernel coming from $L_1$ is defined in $\mathbb{R}^{C \times C \times h_1 \times w_1}$. However, the coupling matrices from the dense layers are defined in $\mathbb{R}^{C \times D_l}$ with $D_l$ representing the number of neurons in each layer. More specifically, the convolutional layer $L_1$ still provides spatially structured information to the phases of the first layer $L_0$. Conversely, the phases of the dense layers affect all the phases of $L_0$ without spatial structure but provide generic information about the identity of the objects. The feedback couplings in the Kuramoto equation do not comprise the global $\epsilon$ desynchronization term (unwarranted since the latter layers are not spatially structured). However, we initialize all the feedback kernels around 0 (before training) to facilitate reaching negative coupling values, thus desynchronizing certain phases of $L_0$ with phases from higher layers when they do not encode for the same object.

## 4.4 Experimental setup

**Datasets.** We perform our experiments on two different datasets. The first one is the Multi-MNIST dataset (Sabour et al., 2017), consisting of images containing two hand-written digits taken from the MNIST dataset. More specifically, we generate empty (black) images of size $32 \times 32$, downsample the MNIST images by a fixed factor (depending on the number of digits we want to fit in the image), and place the first digit at a random location around the upper-left corner. We then randomly pick a distinct second digit and assign it a position in the image under two constraints: not surpassing a predefined maximum amount of overlap, while still fully appearing in the image (none of the digits are cut by the image border). We generate a non-overlapping version of the dataset (maximum overlap = 0%) and an overlapping version where the digits can overlap up to 25% of their active pixels. We use the same procedure to generate images containing more than two digits. Similarly, we generate a version of this dataset with greyscaled CIFAR10 (Krizhevsky et al., 2009) images in the background. This makes the digits less easy to separate for the Kuramoto model and represents a more ecological setting to test our models.

For both of these datasets, the models are evaluated on their ability to recognize and classify the two digits, out of 10 different possible classes. When generating the images, we also generate an associated mask tagging the different objects: first digit and second digit (the same logic potentially extends to a higher number of digits). When the digits overlap, the overlapping region is considered as an additional object. The background is not considered as an object. These masks are used to compute the cluster synchrony loss (Equation 2) and do not provide information about the identity (label) of the digits.

**Baselines.** We compare both versions of our model (KomplexNet and KomplexNet with feedback) with different baselines to highlight our contributions: a real model (with an architecture equivalent to KomplexNets) and a complex model without the Kuramoto synchrony (random phases at the first layer). We also provide comparisons with additional baselines (a real-valued with additional number of parameters and ViT) in Appendix (Figure 12). We additionally show the performance of a complex model with an ideal phase separation as an upper baseline: for this model, using the ground-truth masks, we assign the phases by randomly sampling $N$ equidistant groups on the unitary circle and affecting each value to one object in the image (with $N$ the number of objects, not including the background). When digits overlap, we affect an intermediate phase value (circular mean of the two-digit values) to the overlapping pixels. We also include results and visualizations for a version of KomplexNet with an untrained (fixed) Gaussian coupling kernel to highlight the impact of using spatial masks during training (see Appendix, Figures 16 and 17).

**Model and training.** We train each family of models end-to-end using Adam (Kingma & Ba, 2014), a fixed learning rate of $1e-3$, and a batch size of 128 or 32 depending on the dataset. KomplexNets are trained by accumulating the binary cross-entropy loss at each timestep and then combining it with the synchrony loss from the last timestep. The balance between the two quantities is modulated by a hyperparameter, as illustrated in Equation 7. All experiments are implemented in Pytorch 1.13 (Paszke et al., 2017) and run on a single NVIDIA V100. Each curve in the following plot represents the average and standard deviation over 50 runs with different random initializations. Complementarily, we present in the appendix the test accuracy of the best models on the validation set. To obtain the best hyper-parameter values, we run a hyper-parameter search and use the best combination of values out of 100 simulations. The concerned parameters are: the

desynchronization term $\epsilon$, the gain parameters $\eta_l$ for each layer $l$, the coupling kernel sizes $k_{l_0}$ and $k_{l_1}$, and the balance of the losses $\tau$.

$$L(\hat{y}, y, \theta) = \sum_{t=0}^{T} BCELoss(\hat{y}_t, y) + \tau.CSLoss(\theta_T) \tag{7}$$

**Evaluation.** As we are optimizing two different losses (cluster synchrony and classification), we systematically quantify the performance of our models compared to the baselines along those two separate axes. In all the following sections, we show the first loss under the *synchrony* label and use the *performance* label for the classification objective. The models are all trained with two or three non-overlapping digits. In the next sections, we present the results on *in-distribution* images (two-digit, non-overlapping images, all different from the training set) and then evaluate the *robustness* and *generalization* abilities of the models (without re-training or fine-tuning). We define here robustness as the model's ability to perform the same task given an altered image (compared to the training distribution), while generalization represents the capability of the model to perform a slightly different task from the one it was trained on (here, the categorization of more or less digits than during training).

## 5 Results

### 5.1 In-distribution performance

**Synchrony.** We start by evaluating the ability of the Kuramoto model to correctly separate the objects, compared to the other complex baselines (we cannot explicitly evaluate object separation for the real-valued baseline, as this requires phase information). In Figure 5 panel A, we quantify the quality of the solutions provided by the four models using Equation 2, with the random case as a lower baseline and the ideal case as an upper baseline; we see that the solutions obtained with Kuramoto (and especially with feedback connections) converge over time towards the ideal value. In panel C, we show a qualitative example of phase synchrony for an image taken from the test set (illustrated in panel A). As expected, when the digits are spatially separated on a clean image, the phases obtained with the Kuramoto model almost perfectly synchronize inside the digits and desynchronize between digits: the two clusters have opposite values on the unitary circle (polar plots in the bottom row) and all the pixels within one digit have similar phase values (phase maps across the 8 convolution channels, middle row). In this image, the area where the digits are close and almost touching is assigned a phase value in-between the two clusters; this seems a reasonable solution because the coupling kernel around this region will tend to synchronize the phases across the two digits. This behavior emerges naturally from the network's local coupling dynamics, which promote phase alignment across nearby features (i.e., synchronization of phase variables across network units representing features of the same object). It reflects a smooth, continuous transition in ambiguous regions, rather than a failure of binding. Importantly, this local phase blending aligns with principles from perceptual psychology (Treisman & Gelade, 1980; Treisman, 1998), which recognize that boundaries between overlapping objects can involve graded or competing representations, while still preserving distinct object identities at a global level. As such, this intermediate phase assignment does not impair object-level binding, but rather reflects a biologically plausible resolution of local ambiguity. A smaller coupling kernel would reduce local phase blending by limiting the spatial extent of interactions, but it would then take more timesteps to reach a synchronized state. Visually, the obtained phase maps with KomplexNets seem to lie between the random-phase model (left sub-panel) and the ideal-phase model (right sub-panel) where the clusters show no variance even in ambiguous regions of the image. KomplexNet with feedback shows slightly less dispersed clusters than the feedforward KomplexNet model (middle two sub-panels).

**Performance.** We then evaluate the effect of synchrony on the performance of the model. At each Kuramoto step, we evaluate the KomplexNets and obtain an accuracy curve over time. Conversely, the baselines are deprived of temporal dynamics and therefore yield only one accuracy value. The results are reported in Figure 5, panel B. We can observe that the KomplexNets' performance starts below the real-model and random-phase baselines (because the phases are not synchronized yet) and out-perform them after 5 timesteps

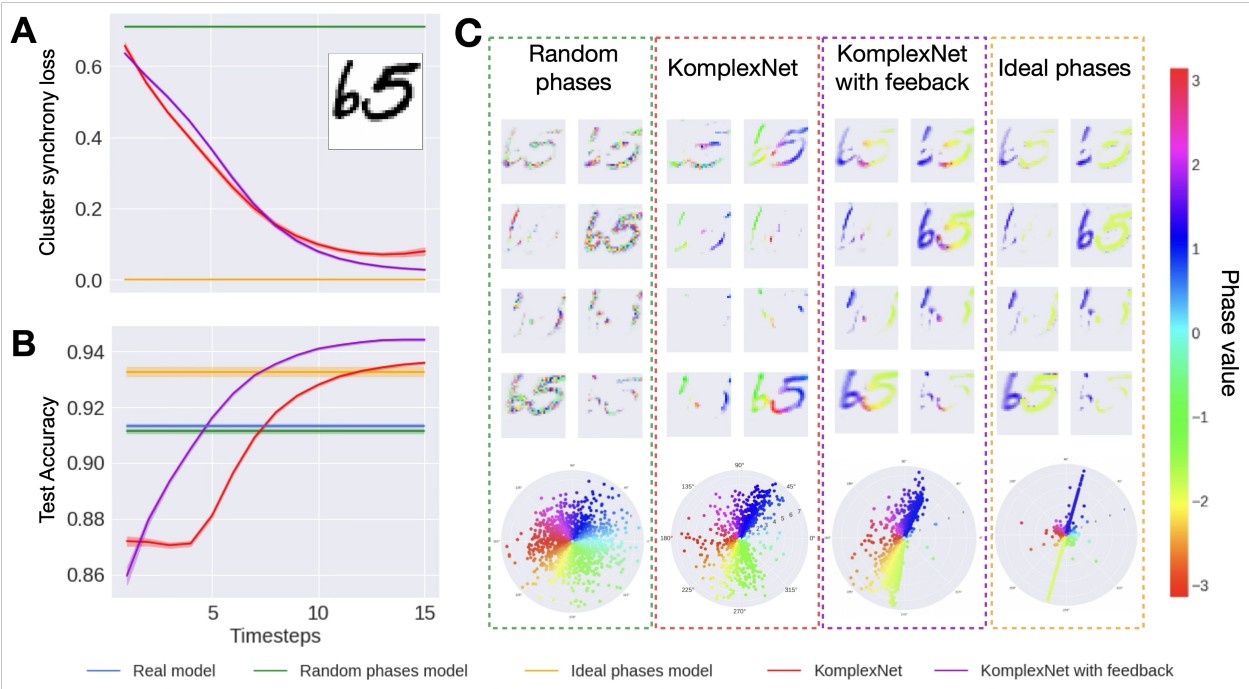

Figure 5: **In distribution results.** Panel A shows the evolution of the cluster synchrony loss through time (computed on the whole test set; lower values indicate better phase separation across digits). Panel B contains the classification performance of KomplexNets compared to the same baselines, as well as the real-valued model. Panel C represents the phases of KomplexNets (red, and KomplexNet with feedback in purple), at the last timestep, compared to the two complex baselines (random phases and ideal phases) on in-distribution images. We show an example of an image (in panel A), the phases of the 8 output channels of the first layer (center sub-panels), and the polar distribution of all the phases - 8 channels combined (bottom sub-panels).

(for KomplexNet with feedback) or 7 timesteps (for KomplexNet without feedback). Surprisingly, after about 10 timesteps, KomplexNet reaches a performance on par with the ideal phase model, while KomplexNet with feedback even outperforms it. This observation sheds light on the phase-synchronization strategy we define as "ideal" here. Because the phase initialization takes place after a convolution, the resulting activity is spread over a slightly larger extent than the "ideal" mask (based on active image pixels). Because of this discrepancy, some activated neurons around the digits are not affected by the phase initialization process, introducing noise in the phase information that is later sent to the rest of the network. Conversely, the Kuramoto dynamic acts on all active neurons, potentially leading to more faithful phase information in the in-distribution case.

We provide in Appendix Figure 10 additional experiments testing the models on more timesteps than during training; the results indicate that the synchronized Kuramoto state is stable over time, and the associated performance improvements persist. We also test in Appendix Figure 13 the effect of the feedback coming from each layer ($L1$, $L2$, $L3$) separately; every single layer provides an amelioration, but the model with feedback from all layers combined shows the greatest performance.

## 5.2 Robustness

We then evaluate the robustness of our trained models on out-of-distribution images. The task remains two-digit classification, and we evaluate both objectives (synchrony and performance) on images with overlapping digits or with additive Gaussian noise.

**Synchrony.** Similarly to the previous section, we observe the cluster synchrony loss of KomplexNet with feedback reaching a lower (i.e. better) score compared to KomplexNet (see Figure 6, first row), for both the "overlap" and the "noise" conditions. In both cases but not surprisingly, the gap with the ideal phases remains higher than before, since the ground-truth phase masks (used for the "ideal-phase" model) are not affected by noise or by digit overlap (when digits overlap, pixels from the overlapping region are considered as a separate group in the ground-truth mask, but this group is not included in the synchrony loss computation). We provide a visualization of the phases for one image (the example shown in Figure 6) in the Appendix Figure 15.

**Performance.** KomplexNet and KomplexNet with feedback show more robustness than the baselines, outperforming the real model in accuracy by 10 to 15% (Figure 6, second row). The real model and the random-phases complex model are very affected by the perturbations, while the ideal-phase complex model shows less altered performance. The accuracy values of KomplexNets lie between the upper baseline and the two other models and surpass them after fewer timesteps than in the previous case (Figure 5), showing how synchronized phases help resolve ambiguous cases. More specifically, KomplexNet with feedback remains more robust than KomplexNet, motivating the use of feedback connections to improve phase synchronization.

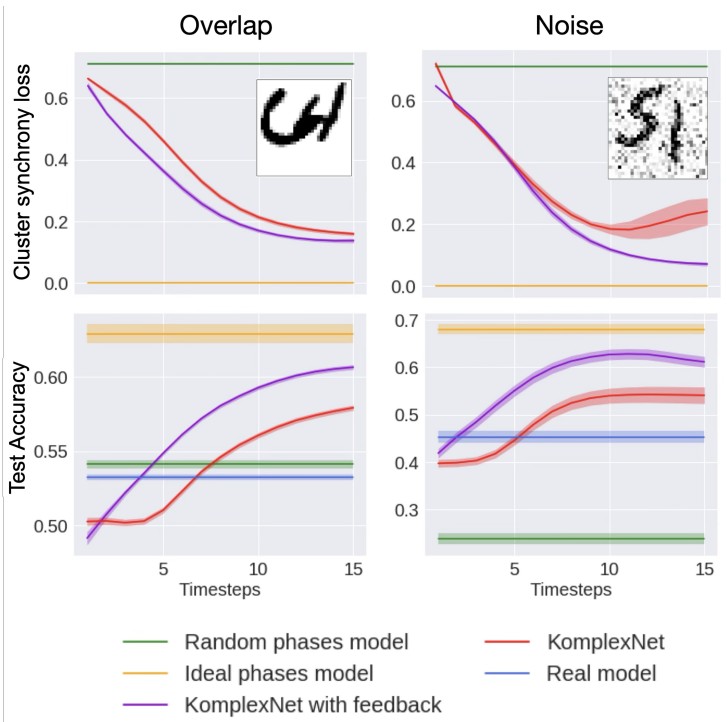

Figure 6: **Robustness performance.** We report the average performance of KomplexNet (red) and KomplexNet with feedback (purple) over time along with the standard deviation for 50 repetitions. We compare it with its real-valued baseline (blue), a complex model with random phase initialization (green), and the ideal phase cluster synchrony (orange). The models are tested on overlapping digits (left column) and noisy images (right column).

### 5.3 Generalization

To evaluate our models' generalization abilities, we report in this section their synchrony and performance when trained on either two or three digits in the images and then tested on the same or a different number of digits, from two to nine digits.

**Synchrony** Figure 7 illustrates the generalization abilities of the KomplexNets at synchronizing the phases in this out-of-distribution setting. Interestingly, despite not having seen 3 digits during training, the coupling kernel of the model trained on two digits can create a third cluster, equidistant to the others on the unitary circle, and correctly affect a single phase value per digit. Likewise, the model trained on three digits adapts well to the two-digit case: the model doesn't create a third phase cluster but only shows higher variance in the two opposite clusters compared to the model trained on two digits.

The evolution of the cluster-synchrony losses illustrates well the general case: losses for both KomplexNets start around the random-phase case and end close to the ideal-phase case. Interestingly, for a given test setting, no matter the training setting (same or different digit number), the models reach approximately the same synchrony values at test time. This observation highlights an additional form of robustness of our models and suggests an object representation ability not present in non-complex and non-synchronized (random-phase) complex models.

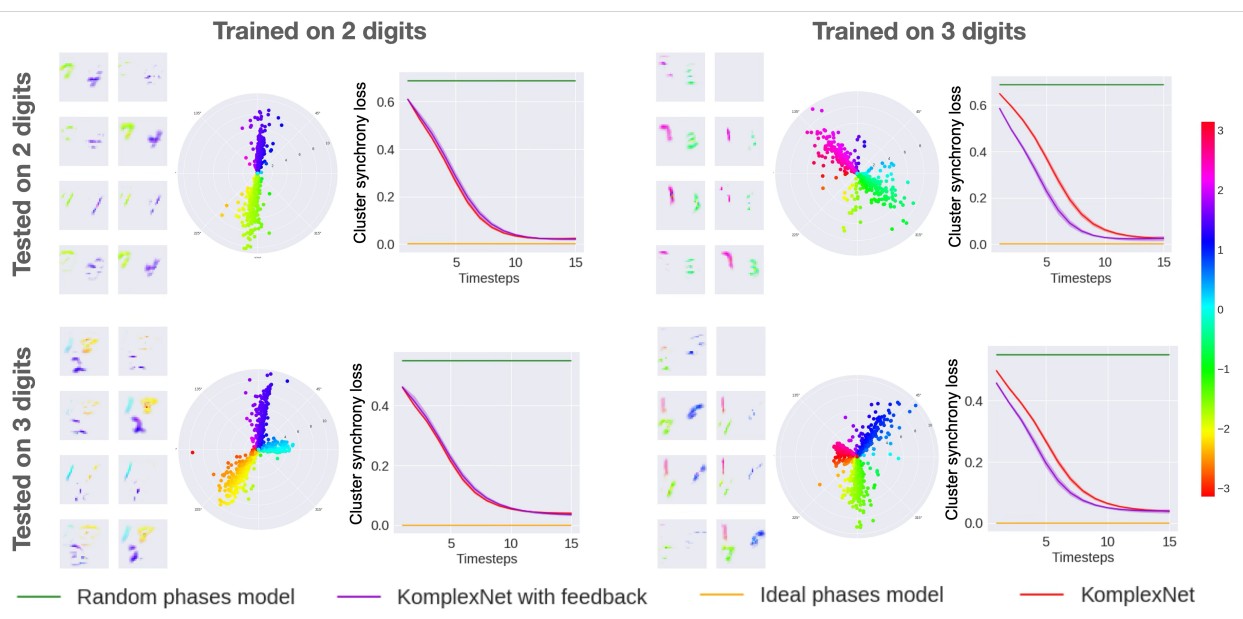

Figure 7: **Generalization to more or less digits.** We show here the generalization ability of KomplexNets trained on two or three digits and tested on two or three digits, evaluated on the synchrony objective. On each panel, we present visualizations of the phases of KomplexNet with feedback on one representative example, at the last timestep, a polar and spatial representation to observe the distribution of the phases and their link with the objects, and the evolution of the cluster synchrony loss through time (over the entire test set), in comparison with the value of the two baselines.

**Performance**   In the same way, we report the classification accuracy of the models and their baselines when trained on two or three digits and tested on two or three digits (Figure 8, panel A). Consistent with the previous results, both KomplexNets outperform the real and random-phases model baselines, both when testing in-distribution and out-of-distribution. More interestingly, both versions of KomplexNet reach almost the same performance on each given test set (within ±3%), no matter the number of digits seen during training. Conversely, the baselines (real and random-phase models) suffer more from this change, leading to a very consequent gap in performance (up to 10%) on the off-diagonal (out-of-distribution testing) plots.

Given this success in generalizing the classification task to one more or one less digit compared to the training set, we next evaluate the maximum number of objects that can be handled by KomplexNets. Consequently, we report the performance of all the models on two to nine digits in the image. In Appendix Figure 14, we report the absolute performance of the models at the last timesteps for each test set (one test set corresponding to a fixed number of digits in the images). As could be expected, performance decreases rapidly as the number of simultaneous objects to classify grows from two or three digits (the number that the models were trained on) up to 9 digits. However, performance drops more quickly for the baselines (real and random-phase models) than for KomplexNets. As the exact advantage of KomplexNets is hard to quantify from this plot, we also report in Figure 8 the difference in performance between all the models and KomplexNet (panel B), and all the models and KomplexNet with feedback (panel C), when trained on either two (first column) or three digits (second column). On these plots, zero difference means that the tested model reaches the same performance as KomplexNet (for panel B, or KomplexNet with feedback for panel C), while a negative difference means that the tested model performed worse than KomplexNet (and conversely). In panel B, we observe a negative difference between KomplexNet and the baselines (except KomplexNet with feedback), persisting up to 8 digits with a peak around 3 to 5 digits. Similarly, KomplexNet with feedback (panel C) systematically outperforms all the other models from two to 8 digits. The nine-digit test set is very hard for all the models because it is the furthest one from the in-distribution case and, as observable in Figure 14, accuracy is very low.

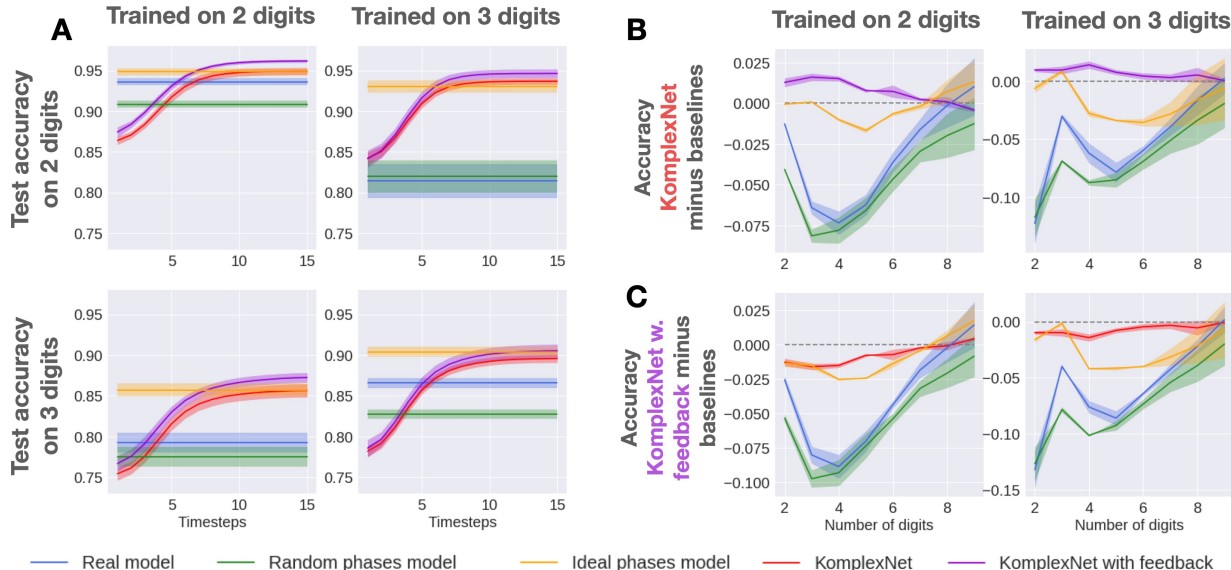

Figure 8: **Performance on generalization.** We report the average performance of KomplexNet (red) and KomplexNet with feedback (purple) over time along with the standard deviation for 50 repetitions. We compare it with its real-valued baseline (blue), a complex model with random phase initialization (green) as well as a complex model with an ideal phase initialization (orange). The models are trained to classify two or three digits and tested on two or three digits (panel A), and up to nine digits (panels B and C). Panel A shows the classification performance per timestep. Panel B and panel C respectively show the difference in performance between KomplexNet and KomplexNet with feedback and the baseline at the last timesteps when tested on different numbers of digits.

Overall, these results reveal that phase synchronization makes the models more robust and general by rendering them less sensitive to out-of-distribution shifts.

## 5.4 Additional experiments

**Experiments on non-uniform background.** To demonstrate that our method is not restricted to images with empty backgrounds, we train our models and the baselines on a version of the multi-MNIST dataset with randomly drawn CIFAR images in the background. The results are presented in Figure 18. Because the meaningful information contained in the image is more difficult to extract, both synchrony and performance measures are affected for all the models. However, KomplexNet and especially KomplexNet with feedback still clearly outperform the baselines. This additional experiment highlights the robustness of the Kuramoto model: the difficulty is in ignoring noisy information in the background, therefore relying less on proximity and more on similarity principles. For this reason, the cluster synchrony loss of KomplexNet is very altered and far from the ideal case, but the feedback connections of KomplexNet with feedback help to bridge the gap.

**Leveraging the temporal dynamic from temporal inputs.** Finally, we evaluate the advantage of the Kuramoto dynamic on dynamic inputs. More specifically, we investigate whether the temporal dimension of the Kuramoto dynamic can act as a memory mechanism when the input is transformed from static images to videos (with digits moving across the frames). We show in Figure 19 the accuracy per timestep relative to the frame with the maximum amount of overlap (denoted timestep 0). Compared to the real baseline, as well as KomplexNets tested on each frame separately, we observe a significant increase in accuracy when the models are tested on the moving object test set. These results suggest that the phase information coming from the previous frames (where the digits overlapped less) is maintained, leading to better performance. Overall, it

confirms the hypothesis that Kuramoto can act as a system with memory, able to use information from the previous timesteps to create a more qualitative phase separation than a system deprived of such context.

## 6 Conclusion

### 6.1 Summary

Here, we propose a model combining complex-valued activations with a Kuramoto phase-synchronization dynamic modeling of the binding-by-synchrony theory in neuroscience. A complex-valued neuron can simultaneously indicate the presence of a feature by its activation amplitude (just as in standard real-valued neural networks), and tag the group or object to which this feature belongs by its activation phase. The Kuramoto model serves as a synchronization process, where the coupling kernels implement an inductive bias of proximity and similarity (akin to Gestalt principles). We show that our model outperforms a real-valued baseline as well as a complex-valued model with random phases on multi-object recognition. More interestingly, the phases play an important role in introducing some notions of object representation in the models, making them more robust to ambiguous cases such as overlapping digits, noisy images, more digits in the images, etc. We additionally propose a way to introduce feedback connections in the model, acting on the phases by using information from higher layers to enhance synchrony. As the phase information becomes more reliable (better cluster synchrony loss) with the feedback extension, the performance, robustness, and generalization of the model increase, highlighting the added value of good phase synchronization for multi-digit classification.

### 6.2 Limitations and discussion

**Model Depth and Scalability.** The models studied in this work are relatively shallow. We intentionally designed a limited architecture to restrict feature representation abilities and highlight the benefits of synchrony in such a setting. While this serves as a proof of concept, scaling up our approach requires identifying tasks and datasets where state-of-the-art models struggle with feature binding in multi-object scenarios. Given the computational demands of training large models on extensive datasets, we opted for a smaller-scale demonstration. Notably, the phase-based grouping in our model is local and parallelizable by design, which suggests favorable properties for scaling to larger architectures and higher-dimensional data. Exploring the scalability of our approach to larger networks and more complex object recognition tasks is a promising direction for future research.

**Hyper-Parameter Dependence.** The Kuramoto dynamics introduced in our model rely on several hyper-parameters, including $\epsilon$, $\eta_l$ for each layer $l$, the coupling kernel sizes $k_{l_0}$ and $k_{l_1}$, and the loss balance parameter $\tau$. Finding optimal values for these parameters requires hyper-parameter tuning to ensure phase synchronization. However, our experiments indicate that these values remain consistent across different tasks (see Table 1), suggesting that hyper-parameter tuning is only necessary when changing datasets.

**Explicit vs. Emergent Synchronization.** A potential limitation of our approach is the explicit addition of Kuramoto dynamics, in contrast to prior work where synchrony emerges through training (Löwe et al., 2022; Stanić et al., 2023). As noted by Stanić et al. (2023), the Complex Autoencoder (CAE) (Löwe et al., 2022) can achieve phase synchronization, but only on simple datasets with a limited number of objects. While Stanić et al. (2023) proposed a method to scale this behavior, they required an additional objective to achieve synchronization on more complex datasets. The emergence of synchrony at scale for complex visual scenes remains an open and non-trivial challenge. Our approach circumvents this difficulty by directly incorporating synchrony via Kuramoto dynamics, providing a controlled framework to assess its benefits. This serves as proof of concept, demonstrating the advantages of complex-valued representations and synchronized activity in visual tasks. Furthermore, our approach aligns with experimental findings in neuroscience (Fries et al., 1997), emphasizing the role of synchrony in early visual processing.

**Neuroscientific Relevance.** Our study is primarily motivated by the *binding by synchrony* theory, a concept widely discussed in neuroscience. While this theory remains influential, it has faced experimental challenges (Roelfsema, 2023; Shadlen & Movshon, 1999). However, recent evidence supports the functional

importance of synchronized activity (Fries, 2023). Additionally, our implementation of feedback connections resonates with the predictive coding framework in neuroscience, where recurrent loops convey top-down predictions that refine lower-level representations (Rao & Ballard, 1999; Friston, 2005). In our model, feedback promotes alignment of phase representations, effectively broadcasting higher-level integration to refine local feature binding. This is consistent with evidence that feedback improves object representation and segmentation, especially under ambiguous conditions. Furthermore, our design parallels physiological findings showing that early visual areas like V1 receive substantial feedback input after initial feedforward sweeps (Lamme & Roelfsema, 2000; Keller et al., 2020). Such feedback has been implicated in refining perceptual organization and enhancing context-dependent interpretation of sensory input, closely aligning with the role of feedback in KomplexNet to improve grouping and segregation over time. Consequently, our work does not aim to provide new insights into biological vision but instead proposes synchrony as a viable solution for feature binding in artificial models. By leveraging neuroscientific principles, we offer an alternative computational mechanism that may inspire future advancements in deep learning architectures.

**Disentangled representations.** KomplexNet introduces an explicit factorization of the representation space by decoupling feature identity (amplitude) from object membership (phase). Traditional deep networks entangle these factors in a single representation space (as proposed by the manifold hypothesis), where natural data form a low-dimensional but tangled subspace (Bengio et al., 2013). In contrast, KomplexNet structures the phase representation explicitly: phase variables encode object binding, while amplitude variables capture feature identity, enabling disentangled representations by design. The Kuramoto dynamics further enforce this separation by structuring the phase space through stable attractor states, as established in dynamical systems theory (Kuramoto, 1975; Breakspear et al., 2010). Each attractor corresponds to an object-centric group of neurons, offering robustness to noise and perturbations (which we empirically confirm and illustrate in Figure 6). Importantly, this synchronization process parallels spectral clustering on a graph: neurons act as nodes, and the learned coupling kernel defines weighted edges based on feature similarity and spatial proximity (Rodrigues et al., 2016). Specifically, the Kuramoto model can be generalized to arbitrary weights between nodes in graph-based Kuramoto networks (see also Equation 12 in (Rodrigues et al., 2016)):

$$\dot{\theta}_i = \omega_i + \sum_j \mathrm{w}_{i,j} \sin(\theta_j - \theta_i) \tag{8}$$

Where $\mathrm{w}_{i,j}$ is the coupling weight between node $i$ and node $j$, and $\omega_i$ is the natural frequency of oscillator $i$ (generally set to 0 because we assume homogeneous or negligible intrinsic frequency). The dynamical system defined by the graph-based Kuramoto model is a gradient flow of the energy function:

$$E(\theta) = -\sum_{i,j} \mathrm{w}_{i,j} \cos(\theta_i - \theta_j) \tag{9}$$

Intuitively, $\cos(\theta_i - \theta_j)$ is minimized when $\theta_i = \theta_j$. So, minimizing $E(\theta)$ leads to synchronization of neighboring nodes with strong weights $\mathrm{w}_{i,j}$. The negative sign ensures that energy is minimized as phases align. This leads to coherent phase clusters that correspond to object groupings (visualized in Figures 3 and 4). This perspective aligns with known connections between synchronization phenomena and graph partitioning methods (Ódor & Kelling, 2019). Moreover, this dynamic phase clustering supports compositional generalization: as observed empirically (Figure 8), KomplexNet generalizes to unseen object counts by flexibly allocating new phase clusters. Theoretically, this aligns with the Kuramoto model's capacity for scalable clustering based on coupling dynamics (Strogatz, 2000), ensuring that object representations remain modular and recombinable. In sum, KomplexNet's explicit use of theoretical properties of the Kuramoto model to act on the phase representation yields grounded benefits for semantic disentanglement and compositionality, leveraging manifold separation and graph-structured dynamics to form robust, interpretable representations.

**More complex and conceptual binding problems.** The binding problem refers to the challenge of associating multiple features belonging to the same object when they are extracted independently, especially in scenes containing multiple objects. Solving this problem involves developing representations that robustly maintain object identity and separation. In this work, we address binding at the level of perceptual grouping, evaluating our solution primarily through object segregation metrics such as the cluster synchrony loss.

However, we view this as a first step toward broader applications, where such phase-based representations could support more abstract forms of reasoning.

Recent work has demonstrated the potential of oscillatory dynamics for solving structured, symbolic problems. For example, Miyato et al. (2024) shows that phase synchronization can encode complex constraint satisfaction tasks, such as Sudoku, by assigning distinct phase states to satisfy relational constraints between variables. Similarly, Muzellec et al. (2024) illustrates how dynamic phase grouping enables robust tracking of object identity over time, even as visual features undergo transformations. These findings suggest that phase synchronization dynamics extend beyond perceptual grouping, offering a flexible mechanism for encoding and maintaining structured, identity-preserving representations that could form the foundation for more abstract reasoning tasks. Moreover, such dynamics could support multi-modal binding – for instance, linking visual features to associated linguistic descriptors (e.g., binding the visual identity of a "red apple" to its corresponding textual or auditory label). By assigning aligned phases to corresponding elements across sensory modalities, phase-based representations could support unified, cross-modal concept formation – enabling grounded semantic representations that dynamically associate visual and language attributes in a compositional and context-sensitive way.

Extending this to conceptual binding, we propose that phase clusters can serve as dynamic labels that flexibly assign features or entities to specific roles in abstract relational structures. For example, in a symbolic reasoning task, one phase cluster could correspond to the role of "subject," another to "predicate," and another to "object." Features or entities participating in these roles would align their phases accordingly, enabling the network to represent not just object identity but also relational structure between objects. Importantly, because phase synchronization is dynamic, these assignments can change flexibly depending on the context or task — a property critical for symbolic reasoning, which requires reusable and composable bindings across different structures (Smolensky, 1990; Hummel & Holyoak, 2003). Thus, dynamic phase assignment provides a potential mechanism for implementing variable-role binding, where the "variable" is the feature or entity, and the "role" is specified by its phase alignment. This dynamic labeling could also extend to predicate-argument binding, such as binding adjectives or attributes (e.g., "red," "tall") to their corresponding referents ("apple," "tree") through shared phase alignment. Recent work (Chaturvedi & Asher, 2024) has shown that transformer-based language models struggle at extracting predicate-argument structure from simple sentences. Phase synchronization across feature spaces could offer a biologically inspired and computationally efficient mechanism for aligning attributes with entities in structured representations. In this way, structured relational representations can emerge from the same oscillatory dynamics that support perceptual grouping in KomplexNet.

Furthermore, recent work on rotating features (Löwe et al., 2024) provides a compelling demonstration of how relational and hierarchical structures can be encoded through learned rotations in feature space. In their model, transformations between entities are represented by applying structured rotations to feature vectors. This concept closely parallels our use of phase representation, where rotation in the complex plane corresponds to shifting relational attributes or grouping features under shared identities. Extending this idea, a natural progression for KomplexNet would be to implement multiple phase manifolds operating at distinct levels (akin to frequencies in the brain). Lowest level phases (i.e., faster oscillations) could encode local perceptual groupings, while highest level phases (i.e., slower oscillations) could capture higher-order relational and conceptual structure. This aligns with several theories in neuroscience that propose oscillatory multiplexing for hierarchical processing. Communication-through-coherence (CTC) (Fries, 2015) theory suggests that synchronized gamma oscillations enable selective communication between brain regions, while the theta–gamma neural code theory (Lisman & Jensen, 2013) posits that fast gamma cycles nested within slower theta rhythms support the concurrent maintenance of multiple items. Both theories offer compatible perspectives on how nested oscillatory dynamics could support complex, multi-level information integration. In parallel, recent work by Jacobs et al. (2025) proposes traveling wave dynamics in recurrent neural networks as a mechanism for spatial integration, where wave propagation accumulates global context over time. While our approach similarly relies on local dynamics to achieve global integration, it differs in using steady-state phase synchronization to support object-centric binding within a single time frame. These perspectives are complementary: traveling waves facilitate broad spatial propagation, while phase alignment enables precise and flexible grouping. Combining these mechanisms in future architectures could enable integration of

information across both space and time, further aligning with principles of cortical computation. By adopting such a multi-frequency and multi-dynamic approach, KomplexNet could not only segregate perceptual features but also build hierarchies of concepts, dynamically linking low-level perception with abstract semantic relationships — a step toward structured, symbolic reasoning within the same dynamical system.

## 7 Acknowledgment

Our work is supported by ONR (N00014-24-1-2026), NSF (IIS-2402875) to T.S. and ERC (ERC Advanced GLOW No. 101096017) to R.V., as well as "OSCI-DEEP" [Joint Collaborative Research in Computational NeuroScience (CRCNS) Agence Nationale Recherche-National Science Fondation (ANR-NSF) Grant to R.V. (ANR-19-NEUC-0004) and T.S. (IIS-1912280)], and the ANR-3IA Artificial and Natural Intelligence Toulouse Institute (ANR-19-PI3A-0004) to R.V. and T.S. Additional support was provided by the Carney Institute for Brain Science and the Center for Computation and Visualization (CCV). We acknowledge the Cloud TPU hardware resources that Google made available via the TensorFlow Research Cloud (TFRC) program as well as computing hardware supported by NIH Office of the Director grant S10OD025181.

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

# A    Appendix

## A.1    Algorithms

We detail here the different steps of both versions of KomplexNet using the pseudo-code algorithm. Algorithm 1 describes the operations of KomplexNet and Algorithm 2 specifies how feedback connections integrate in the previous dynamic.

## A.2    Coupling kernel

---

**Algorithm 1** KomplexNet

---

**Require:** Input image $X$, number of timesteps $T$, set of layers $L_i$ with $i \in 0, ..., N_l - 1$, Kuramoto function $K$, Kuramoto parameters : coupling kernel $R$, desynchrony term $\epsilon$, learning rate $\lambda$

 

    **for** $t \leftarrow 0$ to $T - 1$ **do**
        $a_t \leftarrow L_0(X)$
        $\theta_{0,t} \leftarrow \theta_{0,t-1} + K(a_t, \theta_{0,t-1}, R, \epsilon, \lambda)$
        $z_{0_t} \leftarrow a_t . e^{i.\theta_{0,t}}$
        **for** $i \leftarrow 1$ to $N_l - 1$ **do**
            $z_{l_t} \leftarrow L_i(z_{l-1_t})$
    **return** Predictions $|z_{N_l - 1, T - 1}|$

---

**Algorithm 2** KomplexNet with feedback

---

**Require:** Input image $X$, Initial phases with random values $\theta_{init}$, number of timesteps $T$, set of layers $L_i$, Kuramoto function $K$, Kuramoto parameters : coupling kernels $R_i$, desynchrony term $\epsilon_i$, learning rate $\lambda_i$ with $i \in 0, ..., N_l - 1$

 

    $a_0 \leftarrow L_0(X)$
    $\theta_{0,0} \leftarrow \theta_{init} + K(a_0, \theta_{init}, R_0, \epsilon_0, \lambda_0)$
    $z_{0,0} \leftarrow a_0 . e^{i.\theta_{0;0}}$
    **for** $i \leftarrow 1$ to $N_l - 1$ **do**
        $z_{l,0} \leftarrow L_i(z_{l-1,0})$
    **for** $t \leftarrow 1$ to $T - 1$ **do**
        $a_t \leftarrow L_0(X)$
        $\theta_{0,t} \leftarrow \theta_{0,t-1} + K(a_t, \theta_{0,t-1}, R_0, \epsilon_0, \lambda_0) + \sum_{l=1}^{N_l - 1}[K(|z_{l_{t-1}}|, \theta_{l,t-1}, R_l, \epsilon_l, \lambda_l)]$
        $z_{0,t} \leftarrow a_t . e^{i.\theta_{0,t}}$
        **for** $i \leftarrow 1$ to $N_l - 1$ **do**
            $z_{l,t} \leftarrow L_i(z_{l-1,t})$
    **return** Predictions $|z_{N_l - 1, T - 1}|$

---

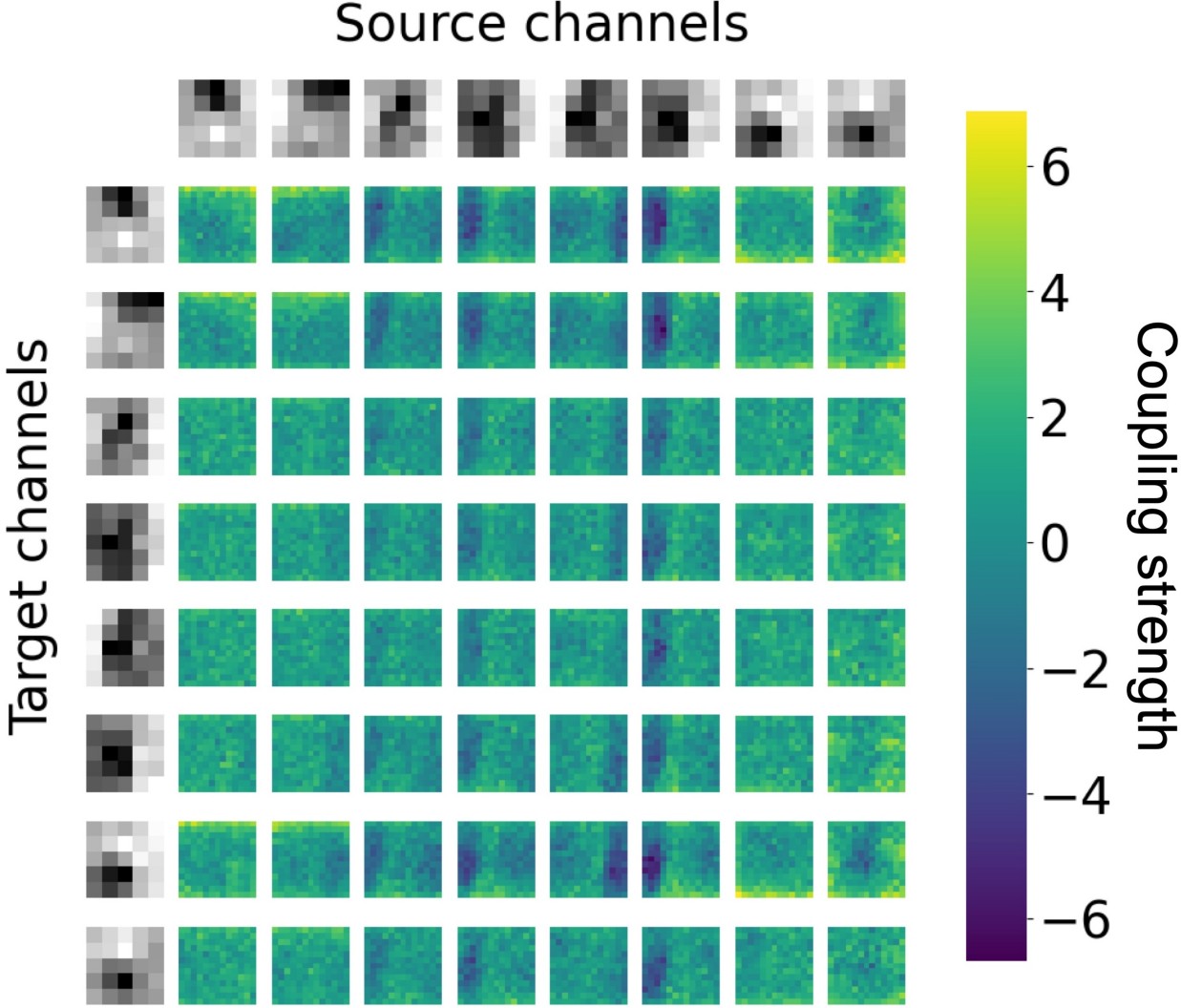

Figure 9: **Visualization of the learned coupling kernel after training.** We show the coupling kernels ($r_{k,cij}$ in Equation 1) learned by KomplexNet, associated with the feature weights learned by the convolution. Each coupling kernel $r_{k,cij}$ represents how phases ($\theta_k$) in the source channel influence phases ($\theta_{cij}$) in the target channel (resulting in a non-symmetric interaction).

### A.3  Complementary results

We show in this section the complementary results and various tests, to give more insights into the functioning of the proposed method.

**Test on more timesteps.**    Without retraining KomplexNets, we test the models on more timesteps than during training. The resulting performance (Figure 10) shows the robustness of the Kuramoto dynamic to maintain a good phase representation over time.

**Best validation model.**    The results presented in the main paper are obtained by averaging the performance (synchrony and accuracy) on 50 different initializations and shown with the standard deviation. Complementary to this choice, we select the best model on the validation set and plot in Figure 11 the synchrony on the first row and the performance on the second row for each type of model and three different test sets (in-distribution,

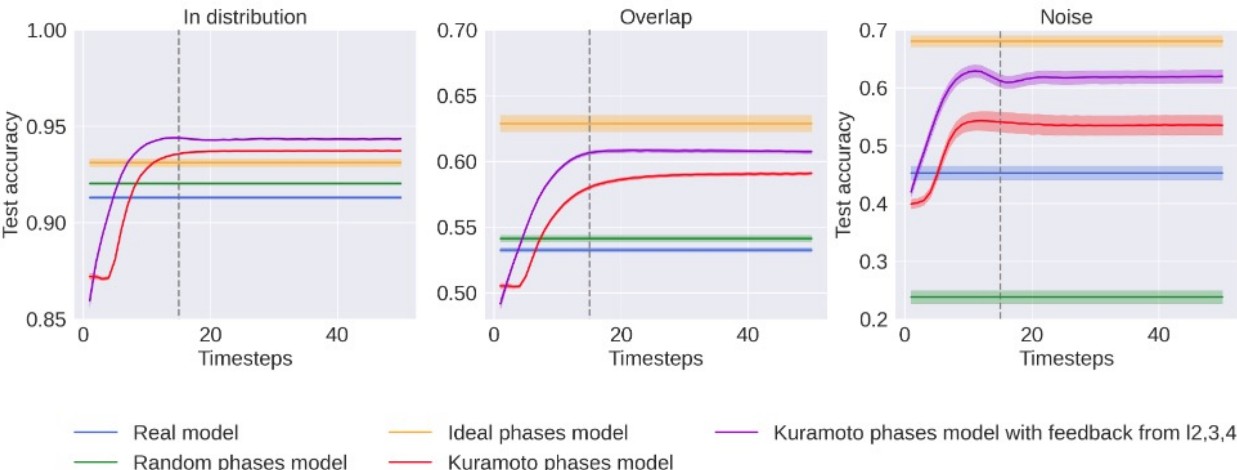

Figure 10: **Results on multi-MNIST when tested on more timesteps.** We report the average performance of KomplexNet (red) and KomplexNet with feedback (purple) over time along with the standard deviation for 50 repetitions. We compare it with a real-valued baseline (blue), a complex model with random phase initialization (green), and the ideal phase cluster synchrony (orange). The models are tested on the in-distribution dataset (left plot), overlapping digits (middle plot), and noisy images (right plot). The vertical bar represents the number of timesteps in the training condition.

overlap, and noise). The results are consistent with the ones in the main text (except for the performance of KomplexNet with feedback on noisy images).

**Extended Benchmark.** We compare both versions of KomplexNet against additional baselines: (1) a real-valued model with increased capacity to match KomplexNet's parameter count, accounting for the additional parameters introduced by the coupling kernel (dotted blue line in Figure 12); and (2) a Vision Transformer (ViT) with a comparable number of parameters, providing a baseline from a different architectural family (brown line in Figure 12. The ViT was trained for 50 epochs—longer than the 30 epochs used for other models—to ensure convergence. While both models perform competitively on the in-distribution test set, they generalize less effectively than KomplexNets when faced with overlapping digits or additive Gaussian noise.

**Influence of the feedback layer.** KomplexNet with feedback receives feedback information from all the layers. To highlight the contribution of each layer specifically, we train separated models with feedback coming from only one of the layers and we report the performance in Figure 13. We can first observe that KomplexNet with feedback (coming from all the layers) outperforms all the other versions on robustness tests, emphasizing the need for various types of information to solve ambiguous cases. Conversely, KomplexNet performs worse than all the models with feedback (both in-distribution and robustness). We additionally provide for each case the value of the hyper-parameters found to maximize the performance in Table 1. In conformance with our expectations, $\epsilon$ has to be higher when the models are provided with feedback connections to compensate for more synchronized activity coming from other layers. The size of the local kernel $k$ remains the same across the versions, corresponding more or less to the size of the digit in the image. Likewise, $k_{l_1}$ is smaller than $k$ to account for the downsampling of $L_1$. Finally, the influence of local synchrony, modulated by $\lambda$ is higher than the influence of the feedback phases ($\lambda_{l_i} \leq \lambda$ for $i \in 1, 2, 3$).

**Raw performance on the generalization test sets.** The main text reports the difference in the performance of KomplexNet and KomplexNet with feedback when tested on two to nine digits to evaluate their generalization abilities. We show in Figure 14 the raw performance of each model and the baselines trained on two or three digits. All the models show more difficulty in classifying correctly as the number of digits in the image increases. However, KomplexNets show consistently a higher performance (see Figure 8).

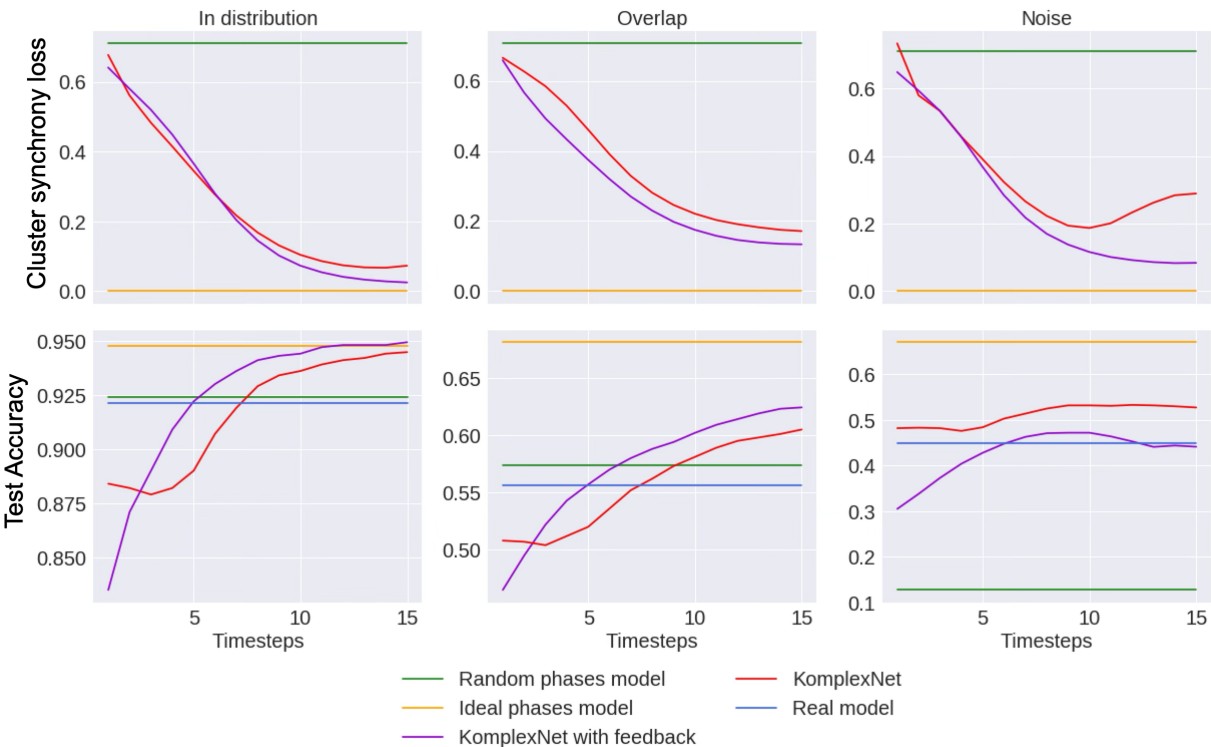

Figure 11: **Results on multi-MNIST.** We report the test synchrony (first row) and performance (second row) of KomplexNet (red) and KomplexNet with feedback (purple) over time along of the best model on the validation set. We compare it with a real-valued baseline (blue), a complex model with random phase initialization (green), and the ideal phase cluster synchrony (orange). The models are tested on the in-distribution dataset (left plot), overlapping digits (middle plot), and noisy images (right plot).

| | $\epsilon$ | $k$ | $k_{l_1}$ | $\lambda$ | $\lambda_{l_1}$ | $\lambda_{l2}$ | $\lambda_{l_3}$ |
|---|---|---|---|---|---|---|---|
| KomplexNet | 0.2 | 13 | - | 0.006 | - | - | - |
| KomplexNet with feedback from l2 | - | 13 | 5 | 0.006 | 0.003 | - | - |
| KomplexNet with feedback from l3 | - | 13 | - | 0.005 | - | 0.004 | - |
| KomplexNet with feedback from l4 | - | 13 | - | 0.005 | - | - | 0.004 |
| KomplexNet with feedback from l2,3,4 | - | 13 | 5 | 0.009 | 0.005 | 0.004 | 0.004 |

Table 1: We perform a hyper-parameter search to find the optimal parameters for the Kuramoto dynamic and report the values for the different versions of the models. $\epsilon$ represents the desynchrony term in the local dynamic, $k$ and $k_{l_1}$ respectively the size of the local coupling kernel and the kernel coming from $L_1$, and $\lambda$ and $\lambda_{l_i}$ for $i \in 1, 2, 3$ modulate the influence of the phase modification by the local dynamic and the one coming from the feedback layers $L_i$ for $i \in 1, 2, 3$.

## A.4 Visualizations

Similarly to the in-distribution tests, we provide in Figure 15 visualizations of the phases of all complex models. The random case remains unchanged. However, we can observe that KomplexNets find a solution for the overlapping digits by almost creating three equidistant clusters, with the overlapping pixels belonging to a cluster in between the two others on the unitary circle. Despite no supervision, KomplexNets found a solution close to the one we adopted for the "ideal" case, consisting of affecting the overlapping pixels to a third cluster, but keeping the two other clusters opposite to each other. Additionally, when we add Gaussian noise to the images, the models struggle more to separate the digits, but KomplexNet with feedback remains able to affect opposite values to the digits (the clusters just show more variance).

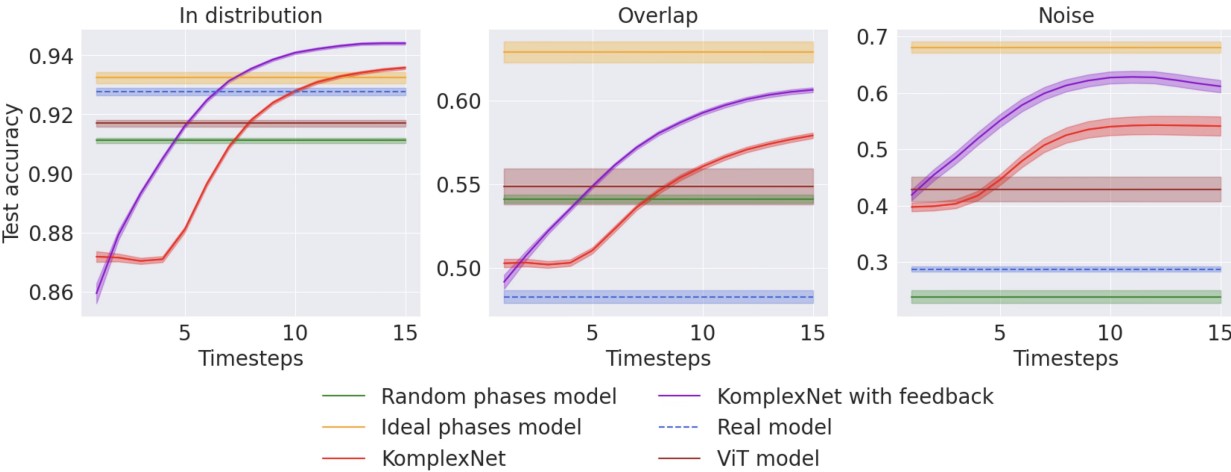

Figure 12: **Extended Benchmark.** We report the test accuracy of two additional baselines: a real-valued baseline accounting for the additional parameters of KomplexNet due to the coupling kernel (dotted blue line) and a ViT to extend the benchmark to different architecture families (brown line). We evaluate these new baselines on the in-distribution test set, overlapping digits and noisy images.

## A.5 Untrained coupling kernel.

While phase synchronization in KomplexNet is driven by end-to-end learning, it primarily emerges from the Kuramoto dynamics independently of gradient-based optimization. Specifically, to demonstrate this, we tested a variant of KomplexNet in which the coupling kernel is defined as a fixed, spatially-local Gaussian and not updated during training. This static, sliding kernel leverages principles of proximity and good continuity, encouraging local phase alignment and desynchronization across more distant locations. As shown in Figure 16, even without learning the kernel, this version of KomplexNet already exhibits meaningful improvements in both test accuracy (Panel A) and phase synchrony (Panel B) over time. However, performance plateaus below that of the fully trained model, highlighting the benefit of learning the coupling kernel jointly with the rest of the network. This underscores the mutually reinforcing relationship between task-driven optimization and synchronization dynamics: synchronization helps learning, and learning sharpens synchronization. To further understand the effects of this fixed-kernel version, we visualize phase distributions and feature activations across layers in Figure 17. The polar plots illustrate the evolution of phase separation from Layer 0 to Layer 3, where color-coded clusters correspond to different objects in the input image. The bottom rows show corresponding feature activations. While phase separation still emerges with the fixed Gaussian kernel, it is less distinct compared to the fully trained version. Together, these results demonstrate that the Kuramoto dynamics alone, when combined with a well-structured coupling kernel, can provide a useful inductive bias for perceptual grouping. Nevertheless, full end-to-end learning yields stronger and more stable synchrony, reinforcing the value of optimizing coupling weights in tandem with task objectives.

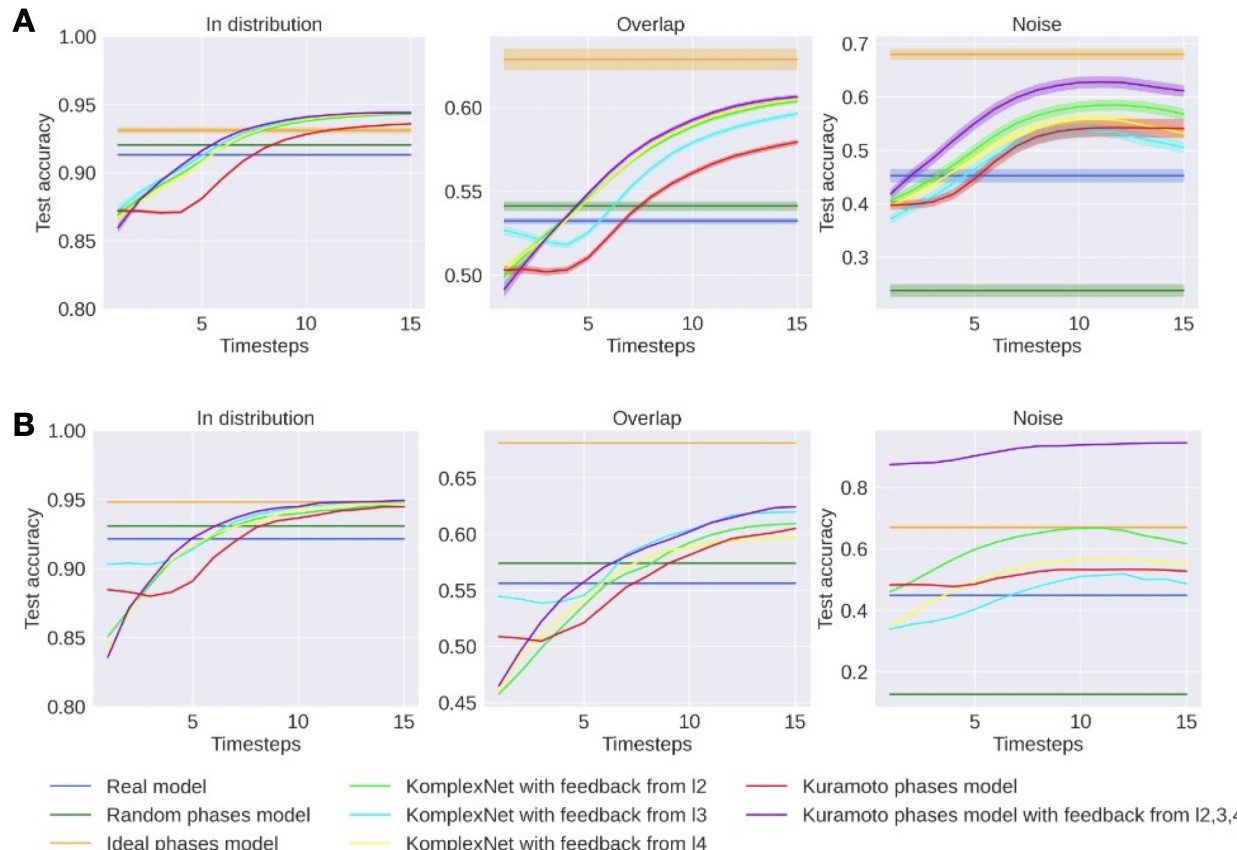

Figure 13: **Performance of Komplexnet with various types of feedback.** We report the average performance of KomplexNet (red) and KomplexNet with feedback (purple) over time along with the standard deviation for 50 repetitions. We additionally show the performance of KomplexNet with feedback from a single layer ($L_2$ in green, $L_3$ in light blue, and $l_4$ in yellow). We compare the models with the usual baselines (real model in dark blue, random phase model in dark green, and ideal phase model in orange). Panel A represents the average and standard deviations over 50 different initializations and Panel B reports the test accuracy of the best initialization on the validation set.

## A.6 Additional experiments

We finally show the results corresponding to the additional experiments detailed in the main text with variants in the datasets to show different uses of KomplexNet.

**CIFAR10 images in the background.** We create an additional dataset with the same digits but with the background filled with RGB content, namely images from the CIFAR10 dataset (see examples in Figure 18). We retrain the models on the new images (the task remains unchanged: 2-digit classification) and report the cluster synchrony loss and accuracy on the test set in Figure 18. Compared to the previous datasets (with a uniform background), the models achieve a lower performance. Indeed, the task is now harder due to less salient information to extract. This version is particularly harder for the Kuramoto model: the propagation of phase synchrony is enhanced by the activated background, even though the information is irrelevant. For this reason, the cluster synchrony is much higher than before and very far from the ideal scenario. As a result, the accuracy is also further from the ideal case. However, KomplexNets are still better than the baselines, confirming that the Kuramoto dynamic remains helpful on RGB images. More specifically, the feedback connections show a clear advantage on the cluster synchrony loss compared to KomplexNet without feedback, resulting in a slight increase in test accuracy.

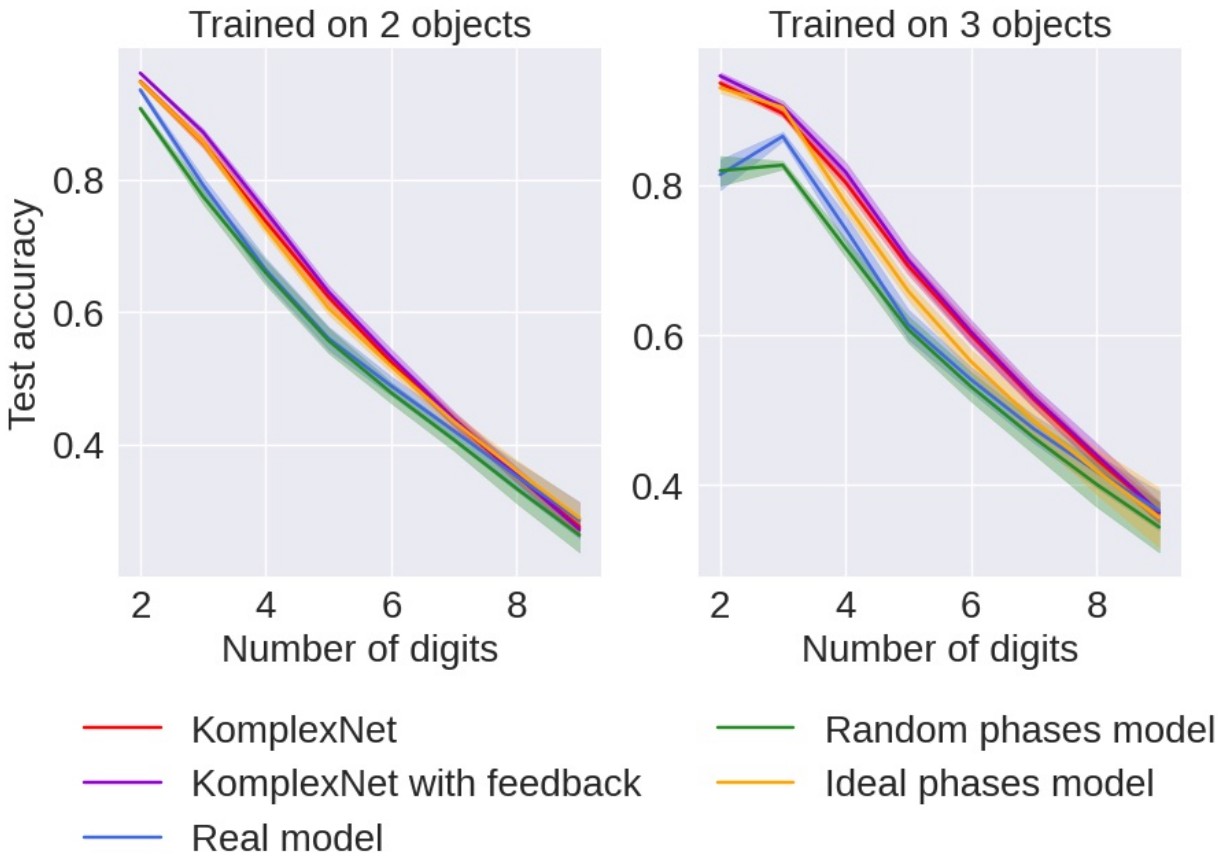

Figure 14: **Results on multi-MNIST with more digits.** We report the average performance of KomplexNet (red) and KomplexNet with feedback (purple) over time along with the standard deviation for 50 repetitions. We compare the models with the usual baselines (real model in dark blue, random phase model in dark green, and ideal phase model in orange). The left plot shows the test performance of the models trained on 2 digits, and the right plot is for the models trained on 3 digits.

**Moving digits.**    The second variant in the dataset is to convert static images into videos with both digits moving across frames. With this version, we aim to evaluate whether we can use the Kuramoto dynamic along with the phase information as a memory mechanism to help resolve ambiguous cases (overlap between digits in this case). To test such a hypothesis, we generate videos of 30 frames, starting from the two digits overlapping (with a maximum of 25% of the active pixels) in the center of the image. We then generate a random trajectory for the first digit and use the opposite trajectory for the second. We let the objects move along these trajectories for 15 frames (making them bounce on the border of the image to prevent them from disappearing and controlling for the amount of overlap at each frame) and use these frames as the first half of the video. We take the opposite trajectories and apply the same procedure to generate the 15 other frames (other half) of the video. In other words, the resulting videos start with both digits at a random location, then slowly move closer to each other, reaching a maximum amount of overlap in the middle before moving away from each other. We are interested in the performance of the model around the middle frame of maximum overlap (10 frames before and after). We use the first frames to let the Kuramoto model converge to a clean phase separation. We evaluate KomplexNet with and without feedback on the videos, as well as the same models tested on each frame separately. For the models tested on the videos, the task has the additional difficulty of adapting the phase information to moving information: each frame corresponds to a single Kuramoto step, potentially leading to noisy artifacts in the phase information because the models were trained on static images. However, the models tested on each frame separately, despite having the

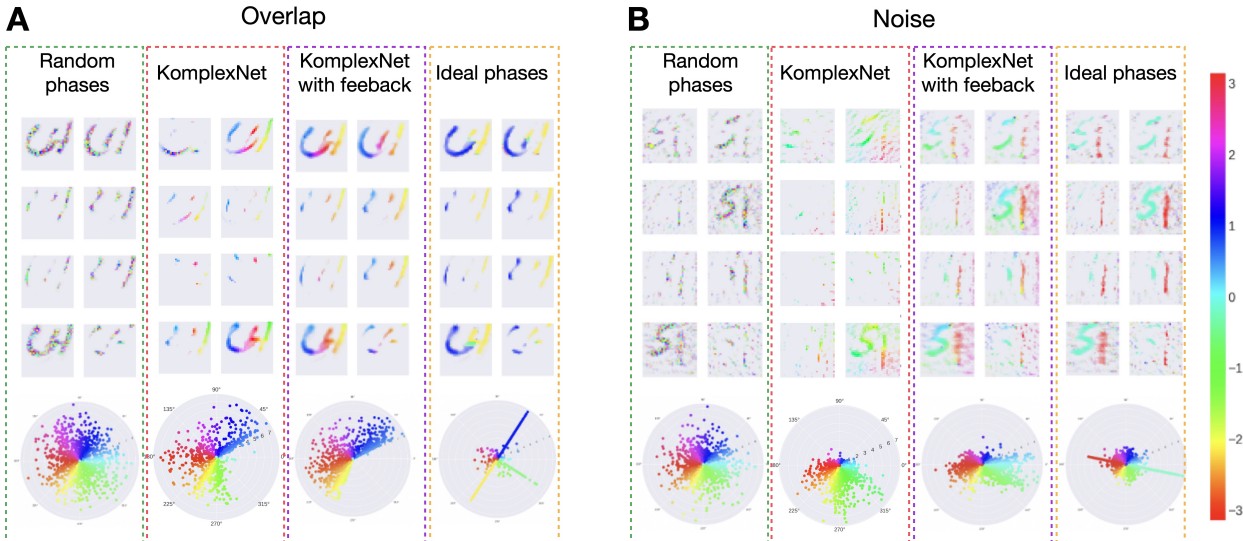

Figure 15: **Robustness on synchrony visualizations.** As in Figure 5, we show the phases of KomplexNet and KomplexNet with feedback, at the last timestep, compared to the two complex baselines (random phases on the left, and ideal phases on the right) on representative examples of overlapping digits (panel A) and additive Gaussian noise (panel B).

time to converge on clean phase separation, do not have access to frames with less overlap between digits. We show in Figure 19 (left panel) the raw performance 10 frames before and after reaching the maximum overlap (denoted frame 0) for KomplexNet with (purple) and without feedback (red) given the two types of input (static in dashed lines and videos in plain lines). We report the performance of the real model (tested on static images only because of the absence of temporal dynamic in the model) to confirm the benefit of KomplexNet on this version as well. We present the videos from Frame 0 to Frame 30 as well as from Frame 30 to Frame 0 to correct for potential bias in the generation of the dataset. On the left Panel, we observe that the models which were presented the moving digits are outperformed by the models tested on single images at the first timesteps. This is explained by the fact that the digits do not overlap (or barely do) 10 frames before Frame 0. Therefore, given several timesteps to converge on the frame, the "static" models reach a cleaner phase separation leading to better accuracy. However, as we get closer to the maximum overlap, the "static" models show a drop in performance, compared to the "dynamic" models, suggesting that the latter manage to make use of the phase separation information from previous frames when the digits did not yet overlap. Finally, as the digits move away from each other, the "static" models recover their performance, outperforming again after a few frames the dynamic model. This result is shown differently on the center panel. We show here the difference in the accuracy of each model with videos running from Frame 0 to Frame 30 relative to the accuracy with videos running backward, from Frame 30 to Frame 0. Every "static" model has a null difference because the dataset was symmetrized. However, the "dynamic" models show a positive difference before Frame 0 and a negative difference after, suggesting that they use the phase information from the previous steps to maintain their phase separation when the digits overlap. However, they take a few steps to recover from the phase corruption induced by the high overlap between digits at the middle frame. Finally, we show on the right panel the difference in performance between static and dynamic models. We can observe a maximum difference at Frame 0 in favor of dynamic models, outperformed by their static real-valued baseline when the amount of overlap is reduced (around frames -10 and 10). These results confirm the possible use of the Kuramoto dynamics as a mechanism of memory, helping resolve extremely overlapping cases and being compatible with dynamical inputs.

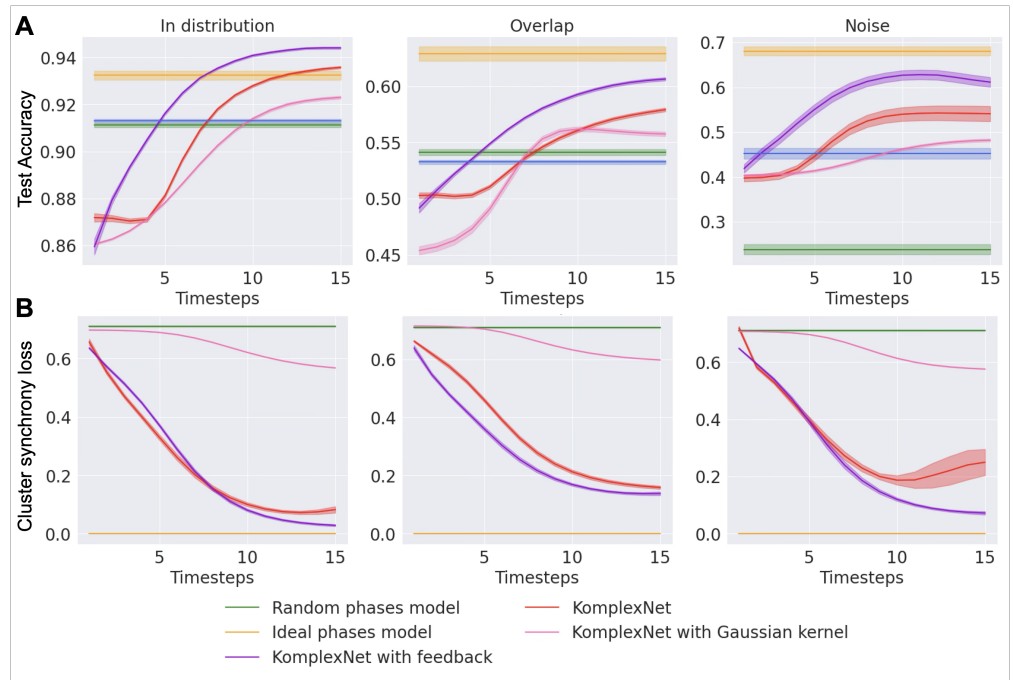

Figure 16: **Results with untrained Gaussian coupling kernel.** We report the Test Accuracy (**A**) and Cluster synchrony Loss (**B**) of KomplexNet with untrained Gaussian coupling kernel (pink), KomplexNet (red), and KomplexNet with feedback (purple) over time for 50 repetitions (mean and standard deviation). We compare it with a real-valued baseline (blue), a complex model with random phase initialization (green), and the ideal phase cluster synchrony (orange).

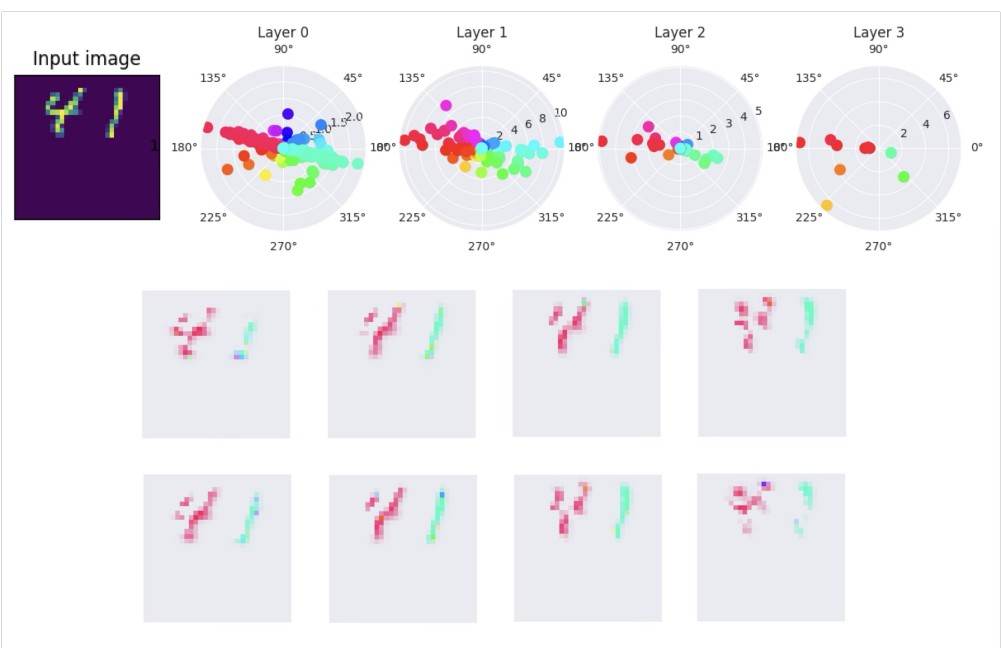

Figure 17: **Visualizations of synchrony in KomplexNet with untrained Gaussian coupling kernel.** As in Figure 4, we show the phases of KomplexNet with an untrained Gaussian coupling kernel, at the last timestep given an input image from the "in-distribution" test set.

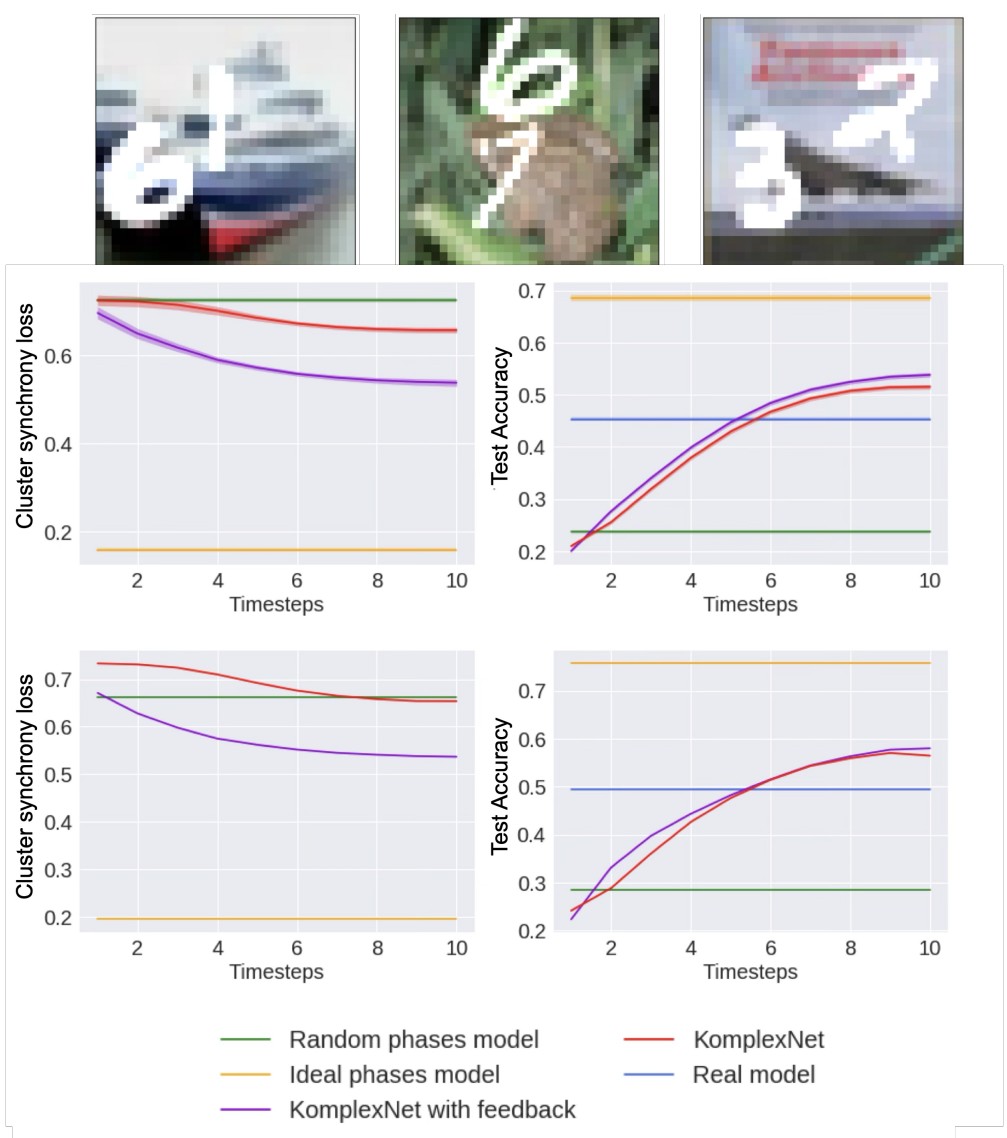

Figure 18: **Results on multi-MNIST with CIFAR in the background.** We provide examples of the stimuli exhibiting multi-mnist digits with a CIFAR image in the background. We report the cluster synchrony loss (first column) and performance (second column) of KomplexNet (red) and KomplexNet with feedback (purple) over time for 50 repetitions (mean and standard deviation in the first row, and best model on the validation set in the second row). We compare it with a real-valued baseline (blue), a complex model with random phase initialization (green), and the ideal phase cluster synchrony (orange).

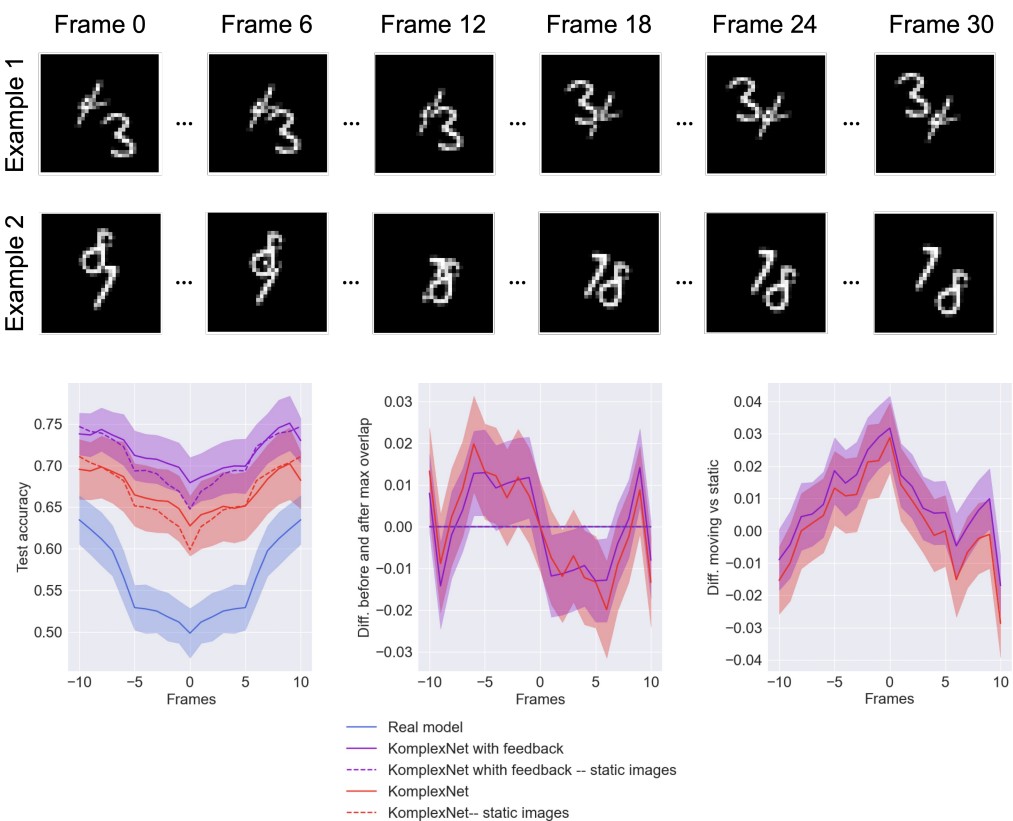

Figure 19: **Results on moving digits.** We provide visualizations of some frames of two example videos. We evenly subsample 6 frames out of the 30 for the sake of visualization. Frame indexing on top is indicated before the normalization with respect to the frame with maximum overlap. We evaluate KomplexNet with (purple) and without feedback (red), as well as a real model (blue) given moving digits and evaluate KomplexNets on static frames (dashed line) versus dynamic videos (plain line). The left panel shows the test accuracy of each model around the frame with the maximum overlap between digits (Frame 0). The middle panel represents the difference in accuracy when tested from Frame 0 to Frame 30 or reversed. The right panel shows the accuracy of the dynamic models versus their static real-valued baseline.

