# OpenReview forum: "Enhancing deep neural networks through complex-valued representations and Kuramoto synchronization dynamics"
_TMLR — Accepted by TMLR_

### Review · Reviewer_HCx7 · 2025-03-05

**Summary Of Contributions:**

This paper presents KomplexNet, an innovative architecture that combines complex-valued neural network representations with Kuramoto synchronization dynamics to enhance the performance of deep learning models on multi-object visual scenes. The authors propose a method where the amplitude component of complex-valued neurons maintains standard CNN functionality, while the phase component facilitates feature binding and scene organization through synchronization. Drawing inspiration from neuroscience's "binding by synchrony" theory, the authors design a system where Kuramoto dynamics drive the phase alignment of features belonging to the same object. The paper evaluates two architectural variants: a feedforward model and a model with feedback connections that further refines phase synchronization through top-down information flow. Through extensive experimentation, the authors demonstrate that both models outperform real-valued counterparts and complex-valued models without Kuramoto synchronization on various tasks including overlapping digit classification, noisy input processing, and out-of-distribution generalization.

**Audience:**

Yes

**Broader Impact Concerns:**

N/A.

**Claims And Evidence:**

Yes

**Requested Changes:**

- The authors can expand the discussion section to analyze how representations formed through phase synchronization differ conceptually from traditional deep learning representations. Include a theoretical analysis of whether phase-synchronized features may naturally afford better semantic disentanglement or compositional properties, referencing existing visualization results rather than conducting new experiments.

- The authors can add a subsection in the discussion that addresses the limitations and potential of phase synchronization for handling more complex conceptual binding problems beyond object segregation. Consider theoretical connections to symbolic reasoning and discuss whether phase synchronization could provide a bridge between perceptual features and abstract concepts.

**Strengths And Weaknesses:**

### Strengths
1.  The integration of Kuramoto dynamics with complex-valued neural networks represents a genuinely innovative approach to the binding problem in artificial neural networks. This cross-disciplinary approach effectively bridges computational neuroscience and deep learning.
2. The paper establishes strong connections to neuroscientific principles, particularly the binding by synchrony theory, providing a biologically-inspired framework for addressing feature binding challenges.
3. The authors conduct thorough evaluations across multiple experimental settings, including in-distribution testing, robustness to noise and overlap, and generalization to varying numbers of objects, demonstrating the versatility of their approach.
4. The visualization and analysis of phase synchronization patterns provide valuable insights into how the model organizes visual information, clearly illustrating the emergence of object-specific phase clusters.

### Weaknesses
1. While the paper demonstrates successful synchronization and classification, it lacks deeper analysis of how the representations learned through phase synchronization differ fundamentally from those in traditional deep learning models. Are these representations more semantically disentangled or compositional?
2. One of the challenge in deep learning involves bridging perceptual features with abstract concepts. The paper does not adequately address whether phase synchronization alone can address more complex conceptual binding beyond simple object segregation.
3. There is limited investigation into how Kuramoto dynamics interact with the learning process itself. How does the model adapt its synchronization strategy during training to accommodate different visual scenes, and how does phase synchronization emerge as a property that interacts with gradient-based learning?

---

> ### Author Response · Authors · 2025-04-19
>
> We appreciate your thoughtful and constructive feedback on our manuscript. We have taken all of your points into careful consideration and provide detailed responses below.
>
> **Theoretical analyses of representations:**
> We have expanded the Discussion (with a new section entitled “Disentangled representations.”) to include a theoretical analysis of how representations learned through phase synchronization differ from those in traditional deep learning models. Specifically, we describe how KomplexNet explicitly factorizes the representation space by separating feature identity (amplitude) from object membership (phase), in contrast to conventional networks where these factors are entangled in a single latent space. We draw connections to the manifold hypothesis and show how Kuramoto-based dynamics structure the phase space via stable attractor states, leading to robust object-centric clusters. Additionally, we highlight formal parallels between phase synchronization and spectral graph clustering, grounding our model in well-established theoretical frameworks. Finally, we argue that this dynamic phase clustering naturally supports compositionality, as the system can flexibly allocate phase clusters to novel object combinations – a claim supported by visualizations included in the manuscript. These additions clarify the semantic benefits of phase-based representations and strengthen the conceptual contribution of our approach.
>
> **Discussion on more complex conceptual binding problems:** We have added a new dedicated subsection to the Discussion  (entitled “More complex and conceptual binding problems.”) that explores how phase synchronization may extend beyond perceptual grouping to support more abstract forms of conceptual binding and symbolic reasoning. In particular, we draw connections to recent work on phase-based models for structured problem solving (e.g., Sudoku solving and object tracking with dynamic phase grouping), and we propose how phase alignment can serve as a flexible mechanism for variable-role binding in relational structures. We also discuss how phase synchronization can support multi-modal binding, such as linking visual and linguistic attributes, and reference recent work on phase-based predicate–argument binding. Furthermore, we situate our proposal within existing neuroscience theories, including both communication-through-coherence (Fries, 2015) and theta–gamma multiplexing (Lisman & Jensen, 2013), to argue that nested oscillatory dynamics could provide a scalable substrate for hierarchical reasoning and concept composition. These additions highlight the potential of phase-based dynamics to bridge low-level perception and high-level symbolic representation within a unified computational framework.
>
> *Fries, P. (2015). Rhythms for cognition: communication through coherence. Neuron, 88(1), 220-235.*
>
> *Lisman, J. E., & Jensen, O. (2013). The theta-gamma neural code. Neuron, 77(6), 1002-1016.*
>
> **Answer to Weakness 3:** In KomplexNet, the coupling strengths between oscillators – which determine the synchronization dynamics – are parameterized and learned via gradient descent, alongside the rest of the network parameters. The Kuramoto dynamics unfold during the forward pass, and their differentiable nature allows gradients to propagate through both amplitude and phase updates. As a result, the network learns to adjust its coupling strategy dynamically based on the visual scene structure, promoting phase alignment for features belonging to the same object while avoiding spurious synchronization. This is due to the combination of the intrinsic properties of the Kuramoto model, combined with the synchrony loss that encourages this behavior.
>
> However, the phase synchronization can also arise from the Kuramoto dynamics independently of gradient-based learning. Specifically, our Kuramoto model, as we designed it through a sliding kernel, leverages principles of proximity and good continuity to synchronize nearby phases and desynchronize distant phases. For instance, when the coupling kernel is initialized as a Gaussian and kept fixed during training, KomplexNet still produces meaningful phase synchrony (see new Figure 16), though not as synchronized as with a learned kernel (see new Figure 15B). The network already benefits from this degree of synchrony at the first layer (see new Figure 15A). Nevertheless, jointly training the coupling kernel with the rest of the network via end-to-end learning further enhances both accuracy and synchrony, highlighting the mutually reinforcing relationship between task optimization and phase alignment. We added these new results in an additional section of the Supplementary Material (Section A.5), but see also the response to Reviewer QqNF.

---

### Review · Reviewer_QqNF · 2025-03-18

**Summary Of Contributions:**

This paper builds on past work using synchronizing neural units and building CNNs with oscillating activity to create a small CNN that uses phase synchronization to segment digits and classify them. This shows the potential value of having a second channel (i.e. the phase) for representing and sharing information across spatial locations. The authors also show that this model can generalize to different numbers of digits than it was trained on, as well as to images with Gaussian noise.

**Audience:**

Yes

**Claims And Evidence:**

Yes

**Requested Changes:**

The authors must address the issue with the real-valued comparison. They can do this either by testing out a more equivalent real-valued network, or by changing the language throughout the text to make it clear that this is not meant to be a real-valued equivalent to KomplexNet, but is just a standard baseline to contextualize the performance.

It would also be good if the authors remarked on the extent to which they think this will translate to larger networks needing to perform more complex object recognition.

Please clarify if in Eqn 3 it is the same weights applied to real and imaginary values.

Eqn 4 is poorly explained. Please use the actual terms from the equation when describing what it does.

Please include example inputs in Fig 15 and 16

**Strengths And Weaknesses:**

Strengths:

Clearly written

The comparisons with and without feedback show an important benefit of those top-down connections

Weaknesses:

The main weakness is that the real-valued network is not a proper comparison to the KomplexNet. In order to call the real-valued network a 'counterpart' to the KomplexNet, it would need to have access to the same training information. As far as I can tell, KomlexNet gets to train its lateral kernel using spatial masks of the digits. The real-valued net does not get such beneficial information and does not have any lateral connectivity at all. In this way it is also not clear how the real-valued network has the same number of parameters as KomplexNet (what about the parameters of the coupling kernel?). There are ways to build real-valued networks that get to make use of spatial masks and lateral interactions, and that would form a more proper comparison. With the results as they are, the authors cannot make a very precise claim about the benefits of their specific model.

A secondary weakness is of course that this is all done on MNIST and it is unclear to what extent this would be useful in any more complex setting. Many proofs of concepts are done on MNIST and then never followed through with in larger networks, raising the question of what they aim to show us in the first place.

 I didn't understand this methodological choice: "In this image, the area where the
digits are close and almost touching is assigned a phase value in-between the two clusters; this seems a
reasonable solution because the coupling kernel around this region will tend to synchronize the phases across
the two digits." In the psychology literature the main benefit of something like binding is that it allows you to represent objects separately and cohesively even if they are overlapping. How important is this choice for the model results?

---

> ### Author Response · Authors · 2025-04-19
>
> We are grateful for your careful reading and insightful suggestions. We have addressed all of your comments in the revised manuscript and provide point-by-point responses below.
>
> **Real-valued comparison:** We agree that the original comparison with the real-valued network could be more precise. We have taken several steps to address this concern:
>
> *Parameter matching:* We revisited the parameter counts and confirmed that the original real-valued baseline had slightly fewer parameters than KomplexNet. To ensure a fairer comparison, we now include a new real-valued baseline matched for parameter count (see new Figure 12). This baseline follows a similar architecture and training setup but is carefully adjusted to match KomplexNet’s capacity more closely. Additionally, the overall conclusions of our previous analyses still hold: the new real-valued baseline performs better on the in-distribution set (but remains below both versions of KomplexNet), and exhibits decreased robustness on the overlap and noise test sets.
>
> *Alternative control without spatial masks:* We acknowledge that we did not implement a real-valued feedforward baseline that leverages ground-truth spatial masks directly, as there is no straightforward or canonical way to do so in a purely feedforward design. Instead, we propose an alternative baseline: a variant of KomplexNet in which the coupling kernel is fixed (e.g., Gaussian) and not trained with supervision from spatial masks. This variant retains the Kuramoto dynamics but removes access to spatial supervision during learning. Results and analysis for this variant are now included in Section A.5 of the Supplementary Material, along with new Figures 16 and 17 (see also response to Reviewer HCx7). In short, this version of KomplexNet already exhibits meaningful improvements in both test accuracy (new Figures 16, Panel A) and phase synchrony (new Figures 16, Panel B) over time. However, performance plateaus below that of the fully trained KomplexNet.
>
> *Clarifying the language in the paper:* To avoid any misleading implications, we have revised the manuscript to more clearly describe the real-valued model as a standard baseline rather than a direct counterpart to KomplexNet. We ensure that all mentions are phrased to reflect their role as a contextualizing reference, not as an equivalent model.
>
> **Scalability:** Complex-valued neural networks are still a relatively novel field, and early models in this space primarily focused on simple datasets and geometric shapes, often without end-to-end training via backpropagation. Our work represents an advance in this trajectory by integrating complex-valued representations with gradient-based optimization and demonstrating their effectiveness on multi-object recognition tasks — a necessary step toward scaling. In parallel, research on complex-valued architectures such as complex-valued autoencoders (Löwe et al., 2022) followed a similar path: initial experiments were performed on simple datasets like basic shapes or single digits, but over time, these models have successfully scaled to handle complex datasets such as natural images and reasoning benchmarks like CLEVR. While we cannot yet guarantee that KomplexNet will scale directly to such levels of complexity, we see this work as an essential foundational step. Notably, the phase-based grouping in our model is local and parallelizable by design, which suggests favorable properties for scaling to larger architectures and higher-dimensional data. Exploring the scalability of our approach to larger networks and more complex object recognition tasks is a promising direction for future research.
>
> We extended the “Model Depth and Scalability” section of the Discussion with these arguments.
>
> *Löwe, S., Lippe, P., Rudolph, M., & Welling, M. (2022). Complex-valued autoencoders for object discovery. Transactions on Machine Learning Research (TMLR).*
>
> **Clarifications Eq. 3:** We confirm that we are applying the same weights to real and imaginary values. We modified the sentence below Eq 3: “The newly obtained activity results from applying the same set of real-valued weights to both the real and imaginary parts of the input activity, modifying the amplitude and the phase jointly.”
>
> (1/2)

---

> ### Author Response · Authors · 2025-04-19
>
> **Clarifications Eq. 4:** We propose to replace the previous explanations with: “Following the formulation first proposed by \citet{reichert2013neuronal}, we apply a gating mechanism that selectively attenuates inputs that are out of phase with the target activity. This prevents a scenario where desynchronized inputs (i.e., those with opposing phases) are treated equivalently to inhibitory inputs resulting from negative weights. Specifically, we compute a modulation term (named classic term in \citep{reichert2013neuronal}) \chi as a function of the input amplitude, and we update the modulation variable m_{z_2}​​ by averaging the previous state m_{z_1}​​ with the newly computed term \chi.
> This averaging serves to smooth the gating dynamics, allowing the network to gradually adapt modulation in response to changes in input amplitude, and ensuring that out-of-phase inputs do not undesirably mimic the effects of inhibition.”
>
> **Example inputs for Figures 15 and 16:** We have added example input images for each dataset in the figures presenting results for CIFAR with background and moving Multi-MNIST. Please note that, due to the inclusion of three new figures in the revision, these examples now appear in Figures 18 and 19, respectively.
>
> **Answer to Weakness 3:** The assignment of an intermediate phase value in regions where digits are close is an emergent property of the learned coupling dynamics, rather than a hard-coded design choice on our part. In areas where features from different objects are spatially adjacent, the coupling kernel promotes local synchrony, naturally leading to transitional phase values between clusters. Importantly, this does not compromise global object separation: despite local blending at the boundaries, the system maintains distinct, coherent phase clusters for each object overall. These boundary effects are spatially limited and do not propagate ambiguity throughout the representation. Moreover, in regions of spatial overlap, there is no uniquely optimal phase assignment – any hard-coded choice would introduce artificial discontinuities – so the system’s tendency to assign an intermediate phase reflects a smooth and reasonable compromise imposed by the dynamics.
>
> Regarding the connection to the psychology literature, we agree that a key role of perceptual binding is to preserve object individuation even under conditions of spatial overlap or ambiguity. Our model is consistent with this principle: although there is some local phase blending at object borders due to smooth coupling dynamics, the model successfully preserves separate and cohesive object representations at the global level. This behavior mirrors findings in biological systems, where neural responses to overlapping stimuli can exhibit local blending or competition at the boundaries, yet still yield clear perceptual segregation at the system level (Roelfsema, 2006; Craft et al., 2007). Moreover, our approach aligns with psychological theories of feature binding, which emphasize maintaining cohesive object representations even in cluttered or overlapping conditions (Treisman & Gelade, 1980; Treisman, 1998).
>
> We have clarified this point in the manuscript to reflect that this intermediate phase behavior is a natural consequence of local dynamics, aligns with principles from perceptual psychology, and does not impair object-level binding.
>
> *Roelfsema, P. R. (2006). Cortical algorithms for perceptual grouping. Annu. Rev. Neurosci., 29(1), 203-227.*
>
> *Craft, E., Schutze, H., Niebur, E., & von der Heydt, R. (2007). A neural model of figure–ground organization. Journal of neurophysiology, 97(6), 4310-4326.*
>
> *Treisman, A. M., & Gelade, G. (1980). A feature-integration theory of attention. Cognitive psychology, 12(1), 97-136.*
>
> *Treisman, A. (1998). Feature binding, attention and object perception. Philosophical Transactions of the Royal Society of London. Series B: Biological Sciences, 353(1373), 1295-1306.*
>
> (2/2)

---

> ### Comment · Reviewer_QqNF · 2025-04-26
>
> I thank the authors for their revisions. I believe they have increased the soundness and clarity of their claims. The work is appropriate for TMLR

---

### Review · Reviewer_tgUR · 2025-04-11

**Summary Of Contributions:**

Authors utilize synchronization across neurons (via Kuramoto model) to design a complex-valued convolutional neural network (with feedback). With ablation experiments (using CNN models as baselines), they show that all three aspects of the model, synchronization, feedback, and complex validness, contribute to the success of the model, i.e., overall test set performance, robustness in the case of perturbed images, and OOD generalization.

**Audience:**

Yes

**Broader Impact Concerns:**

NA.

**Claims And Evidence:**

Yes

**Requested Changes:**

## Major points

- My main concern with this work is the framing of the proposed approach in terms of the Gestalt principle. The use of terms like proximity and similarity may provide intuition for the learned coupling kernels, but this analogy should not be over-interpreted. The claim would be stronger if grounded in contemporary computational models of visual grouping or feature binding; or simply by omitting this association altogether.

- I believe the benchmarking experiments are interesting and convincing, but to my understanding, benchmarked algorithms are all variations of CNN. As such, it remains unclear whether the gains from Kuramoto-based synchronization surpass those achievable with other established approaches. Could you add some models representing traditional baselines as well?

## Minor points

Here are some low level suggestions that could help make improve the presentation

- There is a clear use of jargon throughout the paper. I think authors should aim to define central concepts the first time they come up. For instance, abstract refers to several undefined concepts: object binding, phase alignment, top-down information.

- Could you please comment on the concerns with complex-valued neural networks? I recall there were some issues related to the complex differentiability of loss functions under standard nonlinearities? I think this discussion could be part of the related work section, and I am for once interested in understanding how authors' approach solved existing problems on complex-valued NNs.

- I think the following work could be of interest to include in the discussion: https://arxiv.org/abs/2502.06034? There seems to be some interesting connections between the results of the two papers that would strengthen the appeal of this manuscript to neuroscientists.

- To me, it makes sense that the implementation of the feedback improves the model; which is quite consistent with the predictive coding paradigm in systems neuroscience. Moreover, there is also now clear evidence of feedback received in early brain regions, like V1, once few hundred ms has passed after the first forward pass of the visual information. I would recommend the authors to consider adding a few paragraphs to the discussion about these two topics.

- In Fig. 5A (and later on), when you say "cluster synchrony" decreases, you actually mean loss function values decrease. The synchrony is actually increasing. Could you please present this in a less confusing manner?

**Strengths And Weaknesses:**

## Strengths

- Recently, Kuramoto-type oscillations have gathered quite interest in computer vision, as also noted by the authors. Similarly, on the neuroscience side, the advent of new technologies (e.g., voltage imaging) has enabled studies of traveling waves across brain regions. In that sense, this work is quite timely. Hence, this work is of interest to the TMLR audience.

- Claims are well balanced and the existing literature well cited (though not listed in evaluation)

## Weaknesses

- To my knowledge, Gestalt principle is an outdated concept and its connection to what the networks learn remains largely speculative. Its heavy use in this work is not necessarily providing motivation, rather distracting away from the core contributions.

- The experiments mainly use different variations of the same model, whereas the reader is often left wondering whether how regularly used models in this field (in which I am not an immediate expert) could fit into the picture.

---

> ### Author Response · Authors · 2025-04-14
> **Clarifications**
>
> Thank you for your valuable feedback on our manuscript. While we are working on the rebuttal, could you clarify this point to ensure we address it properly: when you mention that our claim would be stronger if grounded in contemporary computational models of visual grouping or feature binding, which specific models do you have in mind?

---

> > ### Comment · Reviewer_tgUR · 2025-04-14
> >
> > I would like to thank the authors for their very careful read of my review. I believe the key concern I raised centers on the use of Gestalt principles as *the* framing device. While these concepts may provide some intuitive grounding, their strong emphasis in the current manuscript introduces a degree of imprecision that contrasts with the otherwise rigorous and well-executed work. In contemporary neuroscience, the more relevant and empirically grounded echoes of these ideas are found in the study of **brain-wave frequencies**, **traveling waves**, and **local field potentials**, all of which offer a more mechanistic and experimentally tractable framework. Framing the work through these modern concepts would, in my view, better resonate with neuroscience audiences.
> >
> > As a starting point, I recommend the following two recent papers (though I think you can cite more from the LFP literature as well):
> >
> > - A recent theory paper: https://arxiv.org/abs/2502.06034
> > - A recent experimental study: https://www.biorxiv.org/content/10.1101/2024.08.15.607428v1
> >
> > These works highlight how spatiotemporal neural dynamics are now being rigorously examined both theoretically and experimentally. I believe that motivating your synchrony mechanism through such a **directly experimental** lens would not only strengthen the biological relevance of the model but also lead to more precise predictions that one might be able to compare and contrast with modern systems neuroscience.
> >
> > In short, please consider talking about these exciting modern approaches to oscillations and synchrony at least in equal footing as the Gestalt principle, in my preference even with more emphasis. I hope this helps!

---

> > > ### Author Response · Authors · 2025-04-19
> > >
> > > We thank the reviewer for their thoughtful clarification. We greatly appreciate your perspective on aligning our modeling framework with modern experimental findings, which added depth and clarity to our revisions. We have considered each of your points carefully and outline our detailed responses below.
> > >
> > > **Framing around Gestalt principles:** While Gestalt psychology is indeed a historical framework, we do not consider it outdated. In fact, interest in Gestalt principles remains active in the scientific community – a Google Scholar search returns approximately 2,570 papers from the past five years containing the keywords “Gestalt perception” OR “Gestalt psychology”, indicating ongoing relevance in both psychology and neuroscience contexts.
> > >
> > > That said, we fully agree with the reviewer that our work can and should be contextualized within more contemporary systems neuroscience approaches – particularly those focused on neural oscillations, local field potentials (LFPs), and traveling wave dynamics, which offer experimentally grounded and mechanistically precise insights into large-scale coordination and grouping.
> > > In response, we have substantially revised Section 3 of the manuscript. The revised version downplays the historical emphasis on Gestalt principles and instead highlights more recent experimental and theoretical work from systems neuroscience that directly connects to our phase-based modeling approach. We hope this better aligns our contributions with the current landscape of computational neuroscience and makes our framing more relevant to a broader neuroscience audience.
> > >
> > > **Additional baselines:** While we consider CNN-based architectures (such as widely used models like ResNet) to still represent standard traditional baselines, we agree that it is valuable to include models from other architectural families. To this end, we have added a Vision Transformer (ViT) baseline with a comparable number of parameters to KomplexNet (see new Figure 12). This addition extends the benchmark beyond convolutional architectures and provides a stronger basis for evaluating the benefits of Kuramoto-based synchronization relative to alternative approaches. This ViT exhibits better robustness than the real-valued CNN, but still does not match KomplexNet’s performance under challenging conditions with overlapping digits or added Gaussian noise. This supports the added value of phase synchronization beyond what is achievable through attention-based architectures alone.
> > >
> > > **Clarification of central concepts:** We have revised the manuscript to provide clear definitions of all central concepts in the Introduction, including explicit explanations of object binding, phase alignment, and top-down information. In particular:
> > >
> > > *Object binding* refers to the process of associating multiple features extracted from a scene as belonging to the same coherent object.
> > >
> > > *Phase alignment* describes the synchronization of phase variables across network units representing features of the same object, ensuring coherent activation patterns.
> > >
> > > *Top-down information* refers to higher-level contextual signals, derived from global scene understanding or task goals, that influence local processing within the network.
> > >
> > > We have also carefully reviewed the manuscript to minimize jargon and improve accessibility for a broad audience.
> > >
> > > **Complex differentiability:** We agree with the reviewer that prior work on fully complex-valued neural networks — where both activations and weights are complex numbers — has encountered challenges related to differentiability and the design of suitable nonlinearities that preserve complex differentiability (holomorphicity). In our approach, we avoid these issues by using real-valued weights throughout the network, allowing us to perform standard backpropagation without modifications. The complex representation is maintained in the activation space, where we interpret amplitude and phase as distinct functional components: amplitude encodes feature strength, and phase encodes object membership via synchrony.
> > >
> > > To preserve the desirable properties of complex-valued representations, we carefully designed operations in the activation space. Specifically, we apply normalization only to the amplitude, ensuring it remains positive before applying activation functions like ReLU. For the phase component, we intentionally avoid applying normalization or nonlinearities to preserve the geometric interpretation of phase as an angle on the complex circle, which is central to our phase-synchronization mechanism. We clarified this point in the Related Work section of the revised manuscript.
> > >
> > > (1/2)

---

> > > > ### Author Response · Authors · 2025-04-19
> > > >
> > > > **Relevant litterature:** Although this paper was posted on arXiv after our manuscript was submitted, we agree that it is highly relevant to our work and have included a discussion of it in the revised Discussion section (paragraph “Disentangled representations”):
> > > >
> > > > Recent work by~\citet{jacobs2024traveling} proposes traveling wave dynamics in recurrent neural networks as a mechanism for spatial integration, where wave propagation accumulates global context over time. While our approach similarly relies on local dynamics to achieve global integration, it differs in using steady-state phase synchronization to support object-centric binding within a single time frame. These perspectives are complementary: traveling waves facilitate broad spatial propagation, while phase alignment enables precise and flexible grouping. Combining these mechanisms in future architectures could enable integration of information across both space and time, further aligning with principles of cortical computation.
> > > >
> > > > *Jacobs, M., Budzinski, R. C., Muller, L., Ba, D., & Keller, T. A. (2025). Traveling Waves Integrate Spatial Information Into Spectral Representations. arXiv preprint arXiv:2502.06034.*
> > > >
> > > > **Link with feedback in Neuroscience:** We extended the Dicussion section “Neuroscientific Relevance.” with this discussion on the link with our feedback version and the concepts of predictive coding and feedback in neuroscience.
> > > >
> > > > Our implementation of feedback connections resonates with the predictive coding framework in neuroscience, where recurrent loops convey top-down predictions that refine lower-level representations (Rao & Ballard, 1999; Friston, 2005). In our model, feedback promotes alignment of phase representations, effectively broadcasting higher-level integration to refine local feature binding. This is consistent with evidence that feedback improves object representation and segmentation, especially under ambiguous conditions. Furthermore, our design parallels physiological findings showing that early visual areas like V1 receive substantial feedback input after initial feedforward sweeps (Lamme & Roelfsema, 2000; Keller et al., 2020). Such feedback has been implicated in refining perceptual organization and enhancing context-dependent interpretation of sensory input, closely aligning with the role of feedback in KomplexNet to improve grouping and segregation over time.
> > > >
> > > > *Rao, R. P., & Ballard, D. H. (1999). Predictive coding in the visual cortex: a functional interpretation of some extra-classical receptive-field effects. Nature neuroscience, 2(1), 79-87.*
> > > >
> > > > *Friston, K. (2005). A theory of cortical responses. Philosophical transactions of the Royal Society B: Biological sciences, 360(1456), 815-836.*
> > > >
> > > > *Lamme, V. A., & Roelfsema, P. R. (2000). The distinct modes of vision offered by feedforward and recurrent processing. Trends in neurosciences, 23(11), 571-579.*
> > > >
> > > > *Keller, A. J., Roth, M. M., & Scanziani, M. (2020). Feedback generates a second receptive field in neurons of the visual cortex. Nature, 582(7813), 545-549.*
> > > >
> > > > **Clarifications on cluster synchrony loss:** Indeed, when we state that "cluster synchrony decreases", we are referring to the decrease in the value of the *cluster synchrony loss function*. In practice, this reflects an increase in actual synchrony: the variance of phase values within each cluster decreases, while the phase separation between different clusters increases. To avoid confusion, we have revised the manuscript and the figures to clarify this point and consistently refer to the decrease in the cluster synchrony loss rather than "cluster synchrony" itself.
> > > >
> > > > (2/2)

---

> > > > > ### Comment · Reviewer_tgUR · 2025-04-20
> > > > >
> > > > > I thank the authors for their efforts in addressing my and other reviewers' comments. In my opinion, this work is an appropriate addition to TMLR

---

### Author Response · Authors · 2025-04-19
**General Response to Reviewers**

We would like to thank all the reviewers for their thoughtful, constructive, and detailed feedback on our manuscript. All the comments have helped us significantly improve the clarity and framing of the paper.
In response, we have implemented several key revisions throughout the manuscript, as reflected in the updated version submitted with this response.

**Clarified theoretical contributions:** Following Reviewer HCx7 suggestions, we expanded the Discussion section to better differentiate phase-based representations from those in traditional deep learning models. We now also better articulate how our approach supports semantic disentanglement, compositionality, and symbolic binding. We also included a discussion of modern frameworks such as variable-role binding and multi-modal alignment.

**Updated neuroscience framing:** As proposed by Reviewer tgUR, we revised Section 3 to downplay the historical emphasis on Gestalt psychology and instead highlighted more contemporary systems neuroscience frameworks, including oscillations, local field potentials (LFPs), and traveling wave dynamics. We have incorporated references to recent experimental and theoretical work that strengthens the biological grounding of our approach.

**Improved baseline comparisons:** We added two new baselines (suggested by Reviewers QqNF and tgUR) to improve the fairness and breadth of our benchmarking. First, we introduced a real-valued CNN with a matched parameter count to KomplexNet (accounting for the number of parameters of the coupling kernel). Second, we included a Vision Transformer (ViT) baseline to extend comparisons beyond CNN architectures (resulting in a new Figure 12 in Appendix). These additions help contextualize KomplexNet’s performance in a broader architectural landscape.

**Added control experiments:** To isolate the role of spatial masks and learned coupling (to answer questions and concerns from Reviewers tgUR and HCx7), we introduced a version of KomplexNet using a fixed Gaussian coupling kernel (untrained). This control highlights the impact of the spatial supervision used during training and is analyzed in new Figures 16 and 17 (see Section A.5).

**Improved terminology and clarity:** We revised several passages in the main text to clarify our terminology (e.g., "cluster synchrony" vs. "synchrony loss"), better explain local phase blending near object boundaries, and more consistently describe the role and limitations of baselines. Figure captions and section titles were also revised for clarity.

We believe these revisions significantly strengthen the manuscript, and we hope that the updated version better communicates both the motivation and contributions of our work. Again, we thank the reviewers for helping us improve the quality and clarity of the paper.

---

### Decision · Action_Editor_WCfd · 2025-05-16

**Recommendation:** Accept as is

**Comment:**

The main concerns raised in the initial reviews were about the framing of this work on Gestalt principles rather than on more contemporary wave-based neuroscience concepts, lack of non-CNN baselines, the real-valued network baseline not matching the parameter count and use of spatial masks in the proposed KomplexNet, experimental validation in only simple settings, need for deeper analysis of the new representations, and a need to discuss how phase synchronization may facilitate conceptual binding.

The authors have satisfactorily responded to these issues in the rebuttal and the revised paper. The AE thanks the authors and reviewers for their productive discussions that have significantly improved the manuscript.

**Audience:**

The findings of this paper should be of interest to a certain segment of the TMLR audience.

**Claims And Evidence:**

The claims of the paper are supported by convincing evidence after the inclusion of revised and additional baselines during the discussion period. There is agreement among the three reviewers and the AE regarding this.